# Measuring Comprehensive, General Health Literacy in the General Adult Population: The Development and Validation of the HLS_19_-Q12 Instrument in Seventeen Countries

**DOI:** 10.3390/ijerph192114129

**Published:** 2022-10-29

**Authors:** Jürgen M. Pelikan, Thomas Link, Christa Straßmayr, Karin Waldherr, Tobias Alfers, Henrik Bøggild, Robert Griebler, Maria Lopatina, Dominika Mikšová, Marie Germund Nielsen, Sandra Peer, Mitja Vrdelja

**Affiliations:** 1Competence Centre for Health Promotion and Health System, Austrian National Public Health Institute, A-1010 Vienna, Austria; 2HLS_19_ International Coordination Center at the Austrian National Public Health Institute, A-1010 Vienna, Austria; 3Department of Quality Measurement and Patient Survey, Austrian National Public Health Institute, and HLS_19_ International Coordination Center at the Austrian National Public Health Institute, A-1010 Vienna, Austria; 4Ferdinand Porsche FernFH, A-2700 Wiener Neustadt, Austria; 5Department of Education and Psychology, Freie Universität Berlin, 14195 Berlin, Germany; 6Public Health and Epidemiology, Health Science and Technology, Aalborg University, DK-9220 Aalborg, Denmark; 7Unit of Clinical Biostatistics, Aalborg University Hospital, DK-9000 Aalborg, Denmark; 8National Medical Research Center for Therapy and Preventive Medicine, Ministry of Health of the Russian Federation, 101000 Moscow, Russia; 9Clinical Nursing Research Unit, Aalborg University Hospital, DK-9000 Aalborg, Denmark; 10Institute for Water Quality and Resource Management, TU Wien, A-1040 Vienna, Austria; 11National Institute of Public Health, 1000 Ljubljana, Slovenia

**Keywords:** health literacy measurement, instrument development, validation, HLS_19_, HLS_19_-Q12, national adult population

## Abstract

Background: For improving health literacy (HL) by national and international public health policy, measuring population HL by a comprehensive instrument is needed. A short instrument, the HLS_19_-Q12 based on the HLS-EU-Q47, was developed, translated, applied, and validated in 17 countries in the WHO European Region. Methods: For factorial validity/dimensionality, Cronbach alphas, confirmatory factor analysis (CFA), Rasch model (RM), and Partial Credit Model (PCM) were used. For discriminant validity, correlation analysis, and for concurrent predictive validity, linear regression analysis were carried out. Results: The Cronbach alpha coefficients are above 0.7. The fit indices for the single-factor CFAs indicate a good model fit. Some items show differential item functioning in certain country data sets. The regression analyses demonstrate an association of the HLS_19_-Q12 score with social determinants and selected consequences of HL. The HLS_19_-Q12 score correlates sufficiently highly (r ≥ 0.897) with the equivalent score for the HLS_19_-Q47 long form. Conclusions: The HLS_19_-Q12, based on a comprehensive understanding of HL, shows acceptable psychometric and validity characteristics for different languages, country contexts, and methods of data collection, and is suitable for measuring HL in general, national, adult populations. There are also indications for further improvement of the instrument.

## 1. Introduction

International and national policy documents and studies have highlighted the relevance of comprehensive, general health literacy (HL) in general adult populations and recommended measuring and improving it, both by investing in research and by implementing HL policies based on the study results [1,2]. In Europe, the results of the HLS-EU study [3,4] were a driving force for HL agenda setting. The HLS-EU study was conducted from 2009 to 2012 in eight European countries supported by the European Commission. The results demonstrated the relevance of comprehensive, general HL for adult populations concerning public health and health policy. Therefore, the WHO’s report “Health Literacy: The solid facts” [1], which used the theory-based HLS-EU definition of HL and the results of the study, recommend regular, standardized measurement of HL in the general population. Following up on this recommendation, the WHO founded the Action Network on Measuring Population and Organizational Health Literacy (M-POHL) [5,6,7] in 2018, to support the availability of high-quality internationally comparative data on population and organizational HL.

After first publications of the HLS-EU study [3,4,8,9], many countries in Europe and Asia conducted surveys on national population HL, using the HLS-EU study design, partly extending its methodology by including additional instruments and variables as well as more complex analyses of data [10,11]. Since there was little coordination between these studies and their findings were published in individual reports or articles, it was difficult to compare results across countries and the HLS-EU methodology was not developed further in a consensual manner. Therefore, M-POHL considered it important to conduct a multinational standardized study, the Health Literacy Population Survey 2019–2021 (HLS_19_) [12], and by that, to establish a network for regular follow-up surveys. An International Coordination Center was established to provide a study protocol and enable international coordination. In each of the 17 participating countries in the WHO European Region, a National Study Center (NSC) was contracted to conduct the HLS_19_ project: Austria (AT), Belgium (BE), Bulgaria (BG), Czech Republic (CZ), Denmark (DK), France (FR), Germany (DE), Hungary (HU), Ireland (IE), Israel (IL), Italy (IT), Norway (NO), Portugal (PT), Russian Federation (RU), Slovakia (SK), Slovenia (SI), and Switzerland (CH) [12].

The HLS_19_ built on the conceptual framework and definition of comprehensive, general HL developed in the HLS-EU study and its theory-based measurement instruments, the HLS-EU-Q47, the HLS-EU-Q16, the HLS-EU-Q12 and HLS-EU-Q6. HL was defined as “…people’s knowledge, motivation, and competencies to access, understand, appraise and apply information to form judgments and take decisions in terms of healthcare, disease prevention and health promotion to improve quality of life during the life course” [8]. With its definition and concept, the HLS-EU study acknowledged a public health perspective of HL by focusing not only on the patient’s participation in health care but also on people’s prevention and health promotion activities. HL was considered a multidimensional/multifaceted concept [8], which can be illustrated by a 12-cell matrix with four aspects of dealing with health-related information in three domains of health-related tasks (Table 1). The HLS-EU measurement instrument HLS-EU-Q12 operationalized this concept by selecting or creating items for each cell of this 12-cell matrix. By using the self-reported, experienced difficulty of each task as the measurement dimension, the items of the HLS-EU instrument reflect the relational interpretation of HL [3], since the experienced difficulty of a task depends on personal competency as well as on situational demands and resources of the context in which the task is performed.

In the years following HLS-EU, work on the internal differentiation of the comprehensive, general concept and definition of HL and corresponding measurement instruments was continued. It was considered important to follow this trend of differentiation in the HLS_19_ study and to develop and use additional, specific concepts and instruments to measure selected relevant aspects of the comprehensive concept of HL in general adult populations with similar operationalization of their items. Thus, in HLS_19_, the intention was once again to measure general, comprehensive population HL in all participating countries at least by the short form HLS_19_-Q12 but also four specific HL areas as optional packages: (1) digital HL (8 items), (2) communicative HL (with physicians in health care services; long form with 11 items, short form with 6 items), (3) navigational HL (12 items), and (4) vaccination HL (4 items), which are used for validating discriminant validity of HLS_19_-Q12 in this article and are presented in detail in respective chapters of the International Report [12]. The HLS_19_-Q12 builds on the HLS-EU-Q12 but follows the adaptions in wording of response categories and of selected items of measuring comprehensive, general HL in HLS_19_.

The aim of this paper is to examine four research questions concerning HLS_19_-Q12 by using survey data from 17 participating countries of HLS_19_:What is the impact of using dichotomous versus polytomous scoring of HLS_19_-Q12 on its psychometric properties?What are the factorial validity and dimensionality of the two scoring versions of HLS_19_-Q12?How well does HLS_19_-Q12 fulfil aspects, respectively, of content and face validity and of construct validity measured as discriminant validity and concurrent predictive validity?Since HLS_19-_Q12 is offered as a short form of HLS_19-_Q47, how well do the two scoring versions of the short form represent the long form?These research questions will be answered partly by using analyses of dichotomous scored data of different chapters of the HLS_19_ International Report [12] and by new additional analyses using polytomous scored data.

## 2. Materials and Methods

### 2.1. Development of the Instrument for Measuring Comprehensive, General HL in HLS_19_

#### 2.1.1. Development of Its Predecessor, the HLS-EU-Q12

The HLS-EU-Q12 is the short form of the HLS-EU-Q47. Following the original HLS-EU study, HLS-EU-Q47 was used in many follow-up studies for the general population (including all provinces in DE [13,14]), but also for patients, adolescents and students, elderly migrants and asylum seekers, and some other specific subpopulations. Due to the length of HLS-EU-Q47, with 47 items, taking about ten minutes to apply in a personal interview, a demand for shorter forms of the instrument arose. Therefore, the HLS-EU-Q16 was developed based on the data of the 8 original HLS-EU countries [10,11,15,16,17] and further validated for two additional studies in CZ and HU [10], an Austrian study of adolescents [10,18] and an Austrian study on two groups of migrants [10,19,20]. Later studies validating or applying the instruments in further languages were published: in Swedish and Arabic [21], in Somali [22], in Arabi, Dari, and Somali [23], in Italian [24,25], in Indonesian [26], in Austrian German [27], in French [28,29], in Turkish [30], in Japanese [31], in Islandic [32], in Romanian [33], and in an adapted Q18 version in Malaysian [34].

Besides the advantage of being short, though, there are also certain disadvantages of HLS-EU-Q16 (and the HLS-EU-Q6), especially not fulfilling the 12 elements of the underlying theoretical matrix properly. The aim was to develop a shorter version as a subset of the Q47, including as many items of the Q47 as possible, to fulfil the HLS-EU matrix using dichotomous Rasch analyses [35]. That succeeded for 16 items, with the problem of overrepresenting certain cells of the matrix and finding no fitting item for one cell. Furthermore, there was a loss of information by dichotomization of categories and therefore just three instead of four HL levels could be constructed for the scale. Since 16 items were still considered too long for certain research purposes, also a shorter scale, the HLS-EU-Q6, was developed using confirmatory factor analysis (CFA) and was later also validated in the French language [28]. This version represents the matrix of comprehensive, general HL even less well.

Later, independently of the HLS-EU consortium, but using their translated instruments an Asian short form, a 12-item instrument was developed using Taiwanese data [36] and validated for more Asian countries as the HL-SF12 [37] and for people living in rural areas in Vietnam [38]. In addition, a Norwegian HLS-Q12 version was developed and validated [39].

These Q12 short forms differ concerning the methodology used (CFA or Item Response Theory (IRT) modelling), based on data from one country or more countries of different quality and sample size, and the degree of fulfilment of the underlying theoretical model and matrix of comprehensive, general HL.

Therefore, it was decided for HLS_19_ to develop a new short form based on the HLS-EU-Q12 using the original HLS-EU data, with just one indicator in each cell using IRT (Hambleton & Swaminathan, 1985) analyses [40]. Learning from the HLS-EU-Q16 instrument and the later developed short forms (the Norwegian HLS-Q12 and the Taiwanese HL-SF12), a 12-item instrument was developed. On the basis of data from the eight HLS-EU countries, IRT analyses were conducted to achieve maximum overlap with the HLS-EU-Q16 and to identify a 12-item set with the lowest deviance from the assumptions of the Partial Credit Model (PCM; [41]) when analyzed separately for each of the original HLS-EU-countries and two further countries [40]: AT, BG, DE (North Rhine-Westphalia), Greece (GR), IE, Netherlands (NL), Poland (PL), and Spain (ES) (from the HLS-EU study) as well as data from HU and the CZ. It was considered important that just one item represents each cell of the HLS-EU matrix and that items form a locally independent scale (unidimensional and having no response dependency) with an acceptable data-model fit when using the PCM [40]. As such, the 12-item set is not only slightly shorter than the HLS-EU-Q16 but also better represents the underlying model and definition of the HLS-EU instruments.

#### 2.1.2. Adaptation of the HLS-EU-Q12 to the HLS_19_-Q12

Changes in the wording of response categories and selected items were performed for the total HLS_19_-Q47, and by that, also for its related short forms. The original qualifier “fairly” from the 4-point rating scale was removed from the response categories, as it was considered prone to creating differences in translations and interpretations across languages and countries. Item revisions involved both rewording as well as adding or removing examples from individual items. Out of the 12 items of the HLS_19_-Q12, 11 were modified. Criteria for revising the wording of items included: wording which was too complex or difficult to understand (based on expert views and qualitative studies) [39,42,43], the harmonization of similar terms across items (e.g., health and well-being, examples of types of media), indications of difficulties based on quantitative aspects such as high non-response proportions in the HLS-EU, response dependency, item fit (under-discriminating items, differential item functioning (DIF)) of certain items used in other short forms of the HLS-EU-Q47 [39,44,45], and items that do not clearly relate to the use of health-related information. Criteria for removing or adding examples were the use of gender-neutral examples (e.g., removing breast exam), adaptation to societal changes in lifestyle and health-related or medical practices.

The modified instrument was first tested in a focus group study in RU in August 2019 and then field-tested in DE in November 2019. The results of the German field-test were considered for the final English version of the HLS_19_-Q47 and thereby its short form, the HLS_19_-Q12.

#### 2.1.3. Translation Process and Field Testing

The HLS_19_-Q12 instrument was translated by 16 out of the 17 countries (IE used the original English version) into their national language(s). Each NSC organized the translation process. This was mostly done by the data collection agencies and/or other professional translation services. Two forward translations were implemented by ten countries (AT, BE (Dutch translation), CH (German translation), DE, DK, HU, IT, NO, SI, and SK). One forward translation was chosen by countries that cooperated with other countries using the same language (BE for the French translation, CH for the French and Italian translations). Back translation was performed by four countries (IL, NO, RU, and SI). The different translations and, if applicable, additional quality assurance methods for the translated instruments were agreed upon by the HLS_19_ NSCs in the participating countries. All countries, except BG, performed a field test.

### 2.2. Data Collection

From November 2019 until June 2021, the instrument was tested as part of population-based surveys in 17 countries participating in HLS_19_ (Table 2). Countries could choose between collecting data on just the 12 items of the HLS_19-_Q12 or a set of 22 items, which also allows one to calculate the HLS_19_-Q16 short version or the full batterie of the HLS_19_-Q47. Data collection was carried out guided by the HLS_19_ study protocol in the participating countries by national data collection agencies with three exceptions (BG, DK, and SK), where data collection was carried out by the HLS_19_ NSCs. Due to the SARS-CoV-2 pandemic, the surveys vary with respect to data collection method. Telephone-based (CATI), web-based (CAWI), self-administered (SAQ), or face-to-face interviews (PAPI/CAPI) were used. Some countries that had originally planned to use face-to-face interviews switched to CATI or CAWI interviews for pragmatic reasons. Few countries used different survey methods for different sub-populations depending on their accessibility by different data collection methods. Data were collected based on multi-stage random sampling or quota sampling procedures in most countries (Table 2). The data sets were submitted to the HLS_19_ International Coordination Center for data control and creating an English language international data template file for further analysis. Post-stratification weights were applied to improve the estimation of population parameters. For most data sets, the weights were calculated by the HLS_19_ NSCs to best fit the sampling procedure [12].

Since the surveys differed regarding sampling, data collection, language, length of interview, and location of the HL item set in the questionnaire, most analyses were done by the International Coordination Center by country. In some analyses, a reference value (e.g., a mean across all countries weighted equally) is given to facilitate interpretation.

In the HLS_19_ questionnaires, HLS_19_-Q12 was introduced with the following statement: “It is not always easy to get understandable, reliable, and useful information on health-related topics. With the following questions we would like to find out which tasks related to handling health information are more or less easy or difficult. On a scale from very easy to very difficult, how easy would you say it is …” This statement was followed by a set of 12 items:6.… to find out where to get professional help when you are ill? (Instructions: such as doctor, nurse, pharmacist, psychologist)7.… to understand information about what to do in a medical emergency?8.… to judge the advantages and disadvantages of different treatment options?9.… to act on advice from your doctor or pharmacist?10.… to find information on how to handle mental health problems? (Instruction: stress, depression or anxiety)11.… to understand information about recommended health screenings or examinations?12.… to judge if information on unhealthy habits, such as smoking, low physical activity or drinking too much alcohol, are reliable?13.… to decide how you can protect yourself from illness using information from the mass media? (Instructions: e.g., Newspapers, TV or Internet)14.… to find information on healthy lifestyles such as physical exercise, healthy food or nutrition?15.… to understand advice concerning your health from family or friends?16.… to judge how your housing conditions may affect your health and well-being?17.… to make decisions to improve your health and well-being?”

Respondents were asked to choose from one of four response categories: 4 “Very easy”, 3 “Easy”, 2 “Difficult”, 1 “Very difficult”.

### 2.3. Analyses concerning Dimensionality of the Score

#### 2.3.1. Items, Score, and CFA

General data management and the calculation of the score was done in SPSS 27 [46]. Statistical analyses, were conducted using R [47].

The average difficulty (categories “difficult” + “very difficult” combined) for each item by country was visualized as a line chart.

The internal consistency of the scale was measured by the Cronbach’s alpha and the ordinal alpha coefficient. Cronbach’s alpha can be interpreted as lower bound of the true internal consistency [48]. In the literature, a minimum value of 0.7 is recommended [49]. The ordinal alpha based on tetrachoric (for dichotomized items) or polychoric correlation coefficients (for polytomous items can be interpreted as measures for the internal consistence of an item set if they were measured as continuous variables [50,51,52]. The Cronbach’s alpha and the ordinal alpha coefficients were calculated using the psych package in R [53].

Single-factor CFA were conducted as a first check whether the data fits the assumption of a unidimensional one-factor model so that a single score as a measure of general HL can be calculated. This is relevant because of the multifaceted conceptualization of the original HLS_19_-Q47 long form. In addition, the CFAs are expected to test if similar results for the dichotomized and polytomous item sets can be shown. For each country, the following fit indices are given: Standardized Root Mean Square Residual (SRMSR), Root Mean Square Error of Approximation (RMSEA), Comparative Fit Index (CTI), and Tucker–Lewis Index (TLI). The following target values are assumed as indications of a good data model fit [54]:Standardized Root Mean Square Residual (SRMSR) ≤ 0.08Root Mean Square Error of Approximation (RMSEA) ≤ 0.06Tucker-Lewis Index (TLI) ≥ 0.95Comparative Fit Index (CFI) ≥ 0.95

The CFA was conducted using the lavaan package [55] for R [47]. A weighted least square mean and variance adjusted (WLSMV) estimator with diagonally weighted least squares (DWLS) for model parameters is used [49,56,57].

#### 2.3.2. IRT Analyses

For developing the HLS-EU-Q12, as a short form of the HLS-EU-Q47, the Rasch Model (RM) and PCM were used to select the items to be included into the short form (see above). The HLS-EU-Q12 showed fulfilment of Rasch criteria to a certain degree [58]. Therefore, it is of interest to test if the somewhat adapted HLS_19_-Q12, when used in 17 languages and 17 countries, also fulfils Rasch criteria. Knowing this also helps in interpreting the differences found in benchmarking between selected countries. Furthermore, results of RM will identify potentials for further improvement of the instrument in a next round of measuring HL by M-POHL. For these reasons, RM and PCM of HLS_19_-Q12 [58,59] were included in the International Report of HLS_19_ and this article.

The dichotomous RM [35] and its generalization to polytomous items, the PCM [41], possess some favorable properties, namely, sufficiency of raw scores and item marginals as well as specific objectivity (independence of comparisons of persons from the set of items used and vice versa). These properties are not shared by IRT models that do not belong to the family of RMs [60]. Therefore, the data model fit was tested both against the PCM and the RM. For the latter analyses, the answer categories “very easy” and “easy” as well as “very difficult” and “difficult” were merged.

IRT analyses are based on data collected between November 2019 and February 2021 in 15 HLS_19_ participating countries (Table 2). Due to very large sample sizes in some countries (e.g., RU and IE), all analyses were conducted also in a random sample of n = 900 for each of the countries, and PCM analyses on item level were additionally also conducted in four randomly chosen independent subsamples in each of the countries (therefore, the sample sizes in the four subsamples varied according to the total sample sizes in the individual countries between n = 230 and n = 1188). Due to the huge number of significance tests, α = 0.001 was chosen for the individual tests. Analyses were conducted in R using the packages eRm 1.0–1 [61], TAM 3.5–19 [62] and mirt 1.33.2 [63]. For those countries, in which different data collection methods were applied (see Table 2), analyses were conducted on the country level and per data-collection method. This is done only for the IRT analyses to check whether items work slightly differently according to the data-collection method.

Analyses of the overall data-model fit for the PCM included calculation of WLE reliability (Warm’s weighted likelihood estimate) and EAP (expected a posteriori) reliability coefficients [62,64] and calculation of the SRMSR, [65], a global fit statistic based on the comparison of residual correlations of item pairs. Under the assumption of unidimensionality, person parameter estimates from a priori defined subsets of items should not differ significantly [60]. Thus, we applied a procedure of combined principal component analysis (PCA) of residuals and paired *t*-tests [66,67]. Based on PCA of standardized item residuals, we formed two item subsets according to the loadings of the item residuals on the first principal component [66]. Person parameters were estimated in each of the two item subsets and the resulting parameter estimates from the two subsets were compared using paired *t*-tests [66]. Under the assumption of unidimensionality, the proportion of individuals with significantly different person parameters in the two item subsets is small, i.e., ≤5% of the *t*-tests are significant, or the lower bound of a 95% confidence interval (CI) of the observed proportion overlaps 5% [66]. In our analysis, the Agresti–Coull CI was used [68]. We assessed local stochastic independence by means of an adjusted variant of Yen’s Q3 statistic [69] for all item pairs, aQ3, and an effect size of model fit (MADaQ3), which is the average of the absolute values of aQ3 statistics, and *p*-values adjusted according to the Holm procedure [62]. Analyses at item level furthermore included assessing item fit, ordering of response categories and DIF. Item infit statistics and corresponding t-statistics were calculated for the individual items. The expected value of the infit statistic is 1; values > 1 indicate that the item is less predictable than what would be expected according to the IRT model (underfit), values < 1 mean that the item is more predictable than what would be expected according to the expectations of the IRT model (=overfit; [70]). Underfitting items may severely degrade the measurement, whereas overfitting items may overestimate raw score differences [71]. Wright and Linacre [72] suggest that items with infit statistics <0.8 and >1.2 should be eliminated in the construction of a new questionnaire. Smith, Rush, Fallowfield, Velikova, and Sharpe [71] showed that the infit statistic is relatively independent of sample size and there are neglectable differences in the identification of misfit between the cut-off values of 1.2 and 1.3. A central requirement of IRT models is that the item should work invariantly across levels of different person factors, such as gender, education, and health status. We conducted DIF analyses using gender and the dichotomized criteria age (median split) and education (< higher education entrance qualification vs. at least higher education entrance qualification). We conducted a facets analysis in which we set up the criteria as facets (e.g., for gender, item + gender + item × gender) and reran the IRT analysis [62]. The interaction term item*gender yields the DIF magnitude. Furthermore, we applied the Nominal Categories Model to check whether the expected ordering of response categories were supported by the data [73].

For the dichotomous scoring, we conducted Conditional Likelihood-Ratio-Tests (LR-tests; [74]) as global model tests (which simultaneously assess all items regarding DIF) using median test score, education (<higher education entrance qualification vs. at least higher education entrance qualification), median age, and gender as split criteria. A necessary and sufficient condition for a unique solution of the conditional maximum likelihood estimates of the item parameters is well-conditioned data [75]. This means that in every possible partition of the items into two non-empty item subsets at least one person has chosen the answer category 1 on one item in the first subset and answer category 0 on one item in the other subset. Additionally, we calculated individual item-fit statistics (z-statistics [76]), and applied graphical model tests according to Rasch [35], Rasch [77], to examine which items are the source for possible misfit. Furthermore, a global test for local independence, which calculates the sum of absolute deviations between the observed inter-item correlations and the expected correlations [78], was conducted. At the item level, increased correlations between inter-item residuals were checked by means of the *Q3*-statistic [61]. Additionally, item characteristic curve (ICC) plots were used to graphically inspect model fit of the individual items. The ICC plots show how the probability for response category 1 expected by the RM changes with the values of the latent variable. If the deviations of the observed values from the expected values are small, there is close conformity of the data with the model.

### 2.4. Calculation of Aggregate Measures

#### 2.4.1. Calculation of an Overall Score

The raw score can be calculated as a summary measure of the 12 items in two ways, referred to as type D and type P scores:Type D. The score is calculated as the percentage (ranging from 0 to 100) of items with valid responses that were answered with “very easy” or “easy” (i.e., the items were implicitly dichotomized).Type P. The score is calculated as the sum of the item’s numeric values (1 = “very difficult”, 2 = “difficult”, 3 = “easy”, 4 = “very easy”) scaled to a range from 0 to 100.

In either case, the score is calculated only if at least 80% of the items contain valid responses. Otherwise, the score is set to missing.

A score similar to the type D score was already used for the original HLS-EU-Q16 score [18]. The implicit dichotomization has the advantage that it attenuates inequalities due to different extreme response preferences in various subpopulations, which could be beneficial, e.g., for inner national and international comparisons. Under certain conditions, it could also be easier to communicate the meaning of such a score, as it is the mean percentage of items which respondents assessed as “easy” or “very easy”. The disadvantage of the type D score is information loss due to the implicit dichotomization.

A type P score (a sum score scaled to the range of 0 to 50) was already used for the HLS-EU-Q47 score [3].

Higher score values signify a higher level of general HL. A value of 0 denotes the lowest possible and a value of 100 the highest possible level of general HL.

We describe the distributions of these two types of scores by mean, standard deviation, and its quartiles. In addition, density plots by countries are displayed.

#### 2.4.2. Calculation of Levels

Discrete levels of HL that identify respondents with “limited” HL are important in the communication of the results of HL surveys, e.g., Baccolini et al. [79]. We use the following procedures to distinguish between “excellent”, “sufficient”, “problematic”, and “inadequate” levels of general HL. These names for the categorial levels were already used for HLS-EU-Q47.

For the type D score, that is based on the dichotomized items, we propose a procedure that closely reproduces the distribution of the levels based on the type P score but is not derived solely from the score value but also from the percentages of how often certain response categories were selected. For this reason, it is theoretically possible that two respondents with the same type D score value are assigned to different levels of HL. For reasons of comparability and easier comprehensibility, the category labels of the HLS-EU study were retained. These normative labels are defined in a transparent way following a simple ruleset, namely, that these labels should be easy to understand and suggest an intuitive ranking of lower or higher levels of HL. The level of “inadequate”, for example, should be used to describe people for whom most of the tasks included in the HLS_19_-Q12 were “difficult” or “very difficult”, with one task at the most being “very easy”.

The following definitions of cut-off points (as percentages) for the categorial levels of the HLS_19_-Q12 were used (as far as possible based on the HLS-EU study):
Excellent: “very easy” ≥ 50 AND “very difficult” + “difficult” < 8.334For “excellent”, the number of answers with “very easy” should be above ^1^/_2_ and the answers for “very difficult” + “difficult” should be no more than ^1^/_12_.Sufficient: “very easy” + “easy” > 83.33For a level of “sufficient” HL, at least 10 out of the 12 items should be answered with “very easy” or “easy” and not more than 2 out of 12 with “very difficult” or “difficult”.Problematic: all respondents who are not in the groups “excellent”, “sufficient”, or “inadequate” (i.e., once the three other categories have been calculated)The level of “problematic” is the intersecting set of not “excellent”, not “sufficient” and not “inadequate”.Inadequate: “very easy” < 8.334, “very difficult” AND “difficult” ≥ 50For “inadequate”, the number of answers with “very difficult” + “difficult” should be above ^1^/_2_ and for “very easy” should be no more than ^1^/_12_.

For the type P score, we follow the procedure established in the HLS-EU study [3], but adapt it to the range of the score from 0 to 100:Excellent: > 83.33 (i.e., ^10^/_12_ to (incl.) ^12^_/12_)Sufficient: > 66.67 and ≤ 83.33 (i.e., ^8^/_12_ to (incl.) ^10^_/12_)Problematic: > 50 and ≤ 66.67 (i.e., ^6^/_12_ to (incl.) ^8^_/12_)Inadequate: ≤ 50 (i.e., 0 to (incl.) ^6^/_12_)

As in the HLS-EU study, the union of problematic and inadequate levels of general HL is defined as “limited” HL.

### 2.5. Validity

Content, respectively, face validity is judged by evaluating the procedure of selecting indicators for the measurement instrument.

For measuring discriminant validity, we calculate the weighted Pearson correlation coefficients between the HLS_19_-Q12 score and the scores of special health literacies also being developed and applied in the HLS_19_ study: digital HL, communicative HL (with physicians in health care services), navigational HL, and vaccination HL. The correlation coefficients of the scores for special health literacies and the HLS_19_-Q12 score should be high enough (e.g., r > 0.4) to assume they all measure aspects of HL. Yet, since scores for special health literacies are supposed to measure different constructs, they should not be too high to measure somewhat different constructs. One application of using multiple scores for different aspects of HL, is their use within regression models. In regression analysis, a pairwise correlation coefficient of |r| > 0.7 is often suggested as a rule-of-thumb threshold for collinearity [80]. To use the various scores in regression models, the pairwise Pearson correlation is thus aimed to be in the range of 0.4 to 0.7.

To test concurrent predictive validity of the score, we follow the HLS-EU study [3] concerning determinants of HL that there should be a social gradient of HL in computing simple linear regression models with the general HL score as the outcome variable and gender, age, education, self-perceived social status in society, and financial deprivation as predictor variables. As far as consequences of HL are concerned, there are analyses in the HLS_19_ International Report [12] on health behavior, health status, and health care utilization with respect to the dichotomous score. Regarding the polytomous score, we just present in this article the example of simple linear regression models with self-perceived health as the outcome variable and general HL, gender, age, education, self-perceived social status in society, and financial deprivation as predictor variables.

Following the approach of the HLS-EU study [3], the predictors were entered as numeric variables into the regression model. We show the unstandardized b coefficients and the standardized β coefficients as a proxy for relative importance. R^2^, as proportion of the variance in the outcome variable being explained by the predictor variables, is used as a measure for goodness of fit. Following the analyses in the HLS-EU report [3], we focus on the β coefficients. The regression models serve two purposes: (1) Does the general HL score vary with socio-demographic variables, respectively, with self-perceived health in a way comparable to previous studies like HLS-EU? For this reason, we follow the statistical approaches adopted in the HLS-EU study. It should be noted that it is not the goal of this paper to find the best model with HLS_19_-Q12 as outcome variable or as predictor of other outcome variables, but to compare behavior of the variables in comparison with previous research. (2) Do type D and type P scores yield comparable results? The linear models were computed using the R survey package [81].

### 2.6. Extent of Representation of Long form HLS_19_-Q47

Six countries (BG, DE, IE, IT, NO, and SI) participating in HLS_19_ used the HLS_19_-Q47 long-form that contains the items of the HLS_19_-Q12 short form. In order to examine whether the HLS_19_-Q12 is a useable approximation of the HLS_19_-Q47 long form and its subdimensions, we show the weighted Pearson correlation coefficients for the score values for general HL and its conceptual subdimensions [3,8]: (1) Health Care, (2) Disease Prevention, (3) Health Promotion, (4) Access/obtain information relevant for health, (5) Understand information relevant for health, (6) Appraise/judge/evaluate information relevant for health, (7) Apply/use information relevant for health.

## 3. Results

### 3.1. Psychometric Properties

#### 3.1.1. Average Difficulty of the Items

The overall percentage of participants responding “very difficult” or “difficult” varies between 8.1% and 43.0% for the HLS_19_-Q12 items with item 4 (“to act on advice from your doctor or pharmacist”) being the easiest item and item 3 (“to judge the advantages and disadvantages of different treatment options”) being the most difficult. In general, the items in the HLS_19_-Q12 were not rated as predominantly “very difficult” or “difficult”, with the sole exception of DE, where the items 3, 8, and 5 were reported as “very difficult” or “difficult” by 56.1% to 71.2% of the respondents, probably due to somewhat different modes of data collection.

The item difficulties vary by country. The combined percentage of “very difficult” and “difficult” responses ranges from 25.6% (SI) to 71.2% (DE) for the most challenging item 3 (“to judge the advantages and disadvantages of different treatment options”), and from 3.4% (PT) to 17.2% (CZ and SK) for the least difficult item 4 (“to act on advice from your doctor or pharmacist”). Nevertheless, there is a more or less common ranking by difficulty of the tasks across countries (Figure 1).

#### 3.1.2. Non-IRT Analyses

##### Internal Consistency

For the dichotomized items, the values of the Cronbach’s alpha coefficients range from 0.67 (AT) to 0.87 (PT). Except for AT, the values are above the recommended target value of 0.7 (Table 3) [49]. The internal consistency, thus, is acceptable for most countries.

The Cronbach’s alpha coefficient is expected to be lower for dichotomized items than for the original polytomous items. For the polytomous items (4-point rating scale), the values range from 0.8 (DE) to 0.9 (PT, RU).

The values of the ordinal alpha vary by country from 0.84 to 0.94 for both the dichotomized and the polytomous items.

##### Single Factor Confirmatory Factor Analyses

For the dichotomized items, all fit indices (Table 3) indicate a good data model fit. It can thus be concluded that the single factor confirmatory model accounts sufficiently well for the correlation patterns among the HLS_19_-Q12 items.

The items for which the standardized parameter estimates (Appendix A) differ the most across countries are (range ≥ 0.4):
item 4, “to act on advice from your doctor or pharmacist”,item 6, “to understand information about recommended health screenings or examinations”,item 9, “to find information on healthy lifestyles such as physical exercise, healthy food, or nutrition”,item 10, “to understand advice concerning your health from family or friends”.

As these are also among the easiest items (Figure 1), this could cause these discrepancies in the loadings since even small deviances may cause a relatively greater shift.

For the polytomous items (4-point rating scale), the fit indices give a largely similar picture (Appendix A), though for some country data, the RMSEA is above the recommended threshold value of 0.06 [54]. All other fit indices are still within the target range. The range of the standardized parameter estimates (Table 4) is greater than or equal to 0.4, meaning these items contribute differently to the score in different countries, for the following two items:item 6, “to understand information about recommended health screenings or examinations”,item 10, “to understand advice concerning your health from family or friends”.

#### 3.1.3. IRT Analyses

##### Partial Credit Model (PCM) for Polytomous Items

The WLE and EAP reliability coefficients showed acceptable values ≥ 0.79 in all countries (Table 5) and for all data collection methods (Table 6).

According to the PCA/*t*-test procedure, in all countries except NO, the proportion of individuals with significant different person parameters in two item subsets identified by means of PCA exceeds 5% in the random sample of n = 900, and only for NO, the lower bound of the CI overlaps 5% also in the total sample. For AT, BE, CH_CAWI_, CZ, DK, DE, FR, IE, IL, RU, and SI, the lower bound of the CI overlaps 10%. For HU, PT, and SK, the lower bound is > 10% both in the total and the random sample (Table 7). The *SRMSR* statistics are below the cut-off value of 0.08 for good model fit according to [82] in all countries (see Table 8) except for the CATI mode in CH, and for IL also below the more conservative cut-off value of 0.05 [65].

The global test for local independence yielded significant results in all countries both in the total and the random samples. Analyses for the individual item pairs showed that the residuals of four item pairs are significantly correlated in several countries with a residual correlation of r > 0.30 in at least one of the countries (Table 9). One dependent item pair each was observed in the domains healthcare (HC), disease prevention (DP), and health promotion (HP). One dependent item pair was across DP and HP. Nine other item pairs had significant residual correlations r > 0.20 in several countries and further four dependent item pairs were found only for PT.

Table 10 shows the significant results for the item infit statistics in the total samples. Item 8 (“to decide how you can protect yourself from illness using information from the mass media”) had significant infit statistics (*p* < 0.001) in several countries with values of the statistic between 1.15 and 1.35. For IE and SI, the value of the infit statistic is above the cut-off of 1.2, which was suggested by Wright and Linacre [72] for item selection in the development of new questionnaires. All other significant infit statistics indicate overfit of the respective items; however, all infit statistics were > 0.8 (for the infit statistics of all items in the total samples of the different countries see Waldherr, Alfers, and Peer [58]).

Several items are affected by DIF in more than one country both in the total and the random sample of n = 900 and/or at least one of the four independent subsamples. Item 6 (“to understand information about recommended health screenings or examinations”) displayed DIF for age in BE, DK, FR, SI, and CH. For participants older than the median age, in the respective countries, it is relatively easier to understand information about recommended health screenings. In DK and SI, item 6 additionally showed DIF with respect to educational background. Item 9 (“to find information on healthy lifestyles such as physical exercise, healthy food or nutrition”) displayed DIF regarding educational background in AT, CZ, FR, SI, and SK and for age in CZ and SI. Furthermore, we observed DIF for item 1 (“to find out where to get professional help when you are ill”) for gender in AT, for item 3 (“to judge the advantages and disadvantages of different treatment options”) for education in AT and CH_CATI_, for item 5 (“to find information on how to handle mental health problems”) for age in IE, for item 8 (“to decide how you can protect yourself from illness using information from the mass media”) for education in AT and SI, for item 10 (“to understand advice concerning your health from family or friends”) for education in DK, and for item 11 (“to judge how your housing conditions may affect your health and well-being”) for age in DK, IE and NO. In DE, HU, IL, PT, and RU, none of the items had a significant z-statistic for the split criteria used [58].

Applying the Nominal Categories Model to check the empirical ordering of the response categories revealed unordered response categories both in the total and the random samples for item 1 (“to find out where to get professional help when you are ill”) in DE, FR, HU, and IE, item 2 (“to understand information about what to do in a medical emergency”) in FR and IE, item 3 (“to judge the advantages and disadvantages of different treatment options”) in IE and NO, item 4 (“to act on advice from your doctor or pharmacist”) in FR and HU, item 5 (“to find information on how to handle mental health problems”) in HU, item 6 in DE, FR, and IE, item 8 (“to decide how you can protect yourself from illness using information from the mass media”) in IE, item 9 (“to find information on healthy lifestyles such as physical exercise, healthy food or nutrition”) in IE, item 10 (“to understand advice concerning your health from family or friends”) in DE, FR, and IE, and item 11 (“to judge how your housing conditions may affect your health and well-being”) in AT and IE. In the Portuguese data, all items had unordered response categories, and in the Irish data, 11 items had unordered response categories. Closer inspection showed that in PT, at least two third of the persons have chosen the answer category “easy” in all items (e.g., for item 10, approximately 84%). On the contrary, the response category “very difficult” was very rarely chosen (by < 1% of persons) in some items in several countries. The frequency distribution for answer category “very difficult“ varies in PT from 0.30% (item 4) to 13.80% (item 2), for “difficult“ from 3.15% (item 4) to 57.31% (item 3), for “easy“ from 25.06% (item 3) to 83.95% (item 10), and for “very easy“ from 3.82% (item 3) to 67.89% (item 4). This could be the reason for the large number of items with unordered response categories, since in the case of low endorsement rates, in some of the categories, the estimation of the parameters of the Nominal Categories Model may be affected such that response categories are tagged as unordered [83]. In IL, RU, SI, and SK, no unordered response categories were observed in the random samples.

##### Rasch Model

The likelihood ratio (LR)-test with median score as split criterium yielded significant results (*p* < 0.001) in the random samples of n = 900 in DE and IE. In AT, PT, RU, and SI, this test could not be conducted due to ill-conditioned data. The LR-tests with split criterium median age were significant in BE, CH, DK, FR, NO, PT, and RU. For gender and educational background, no significant LR-tests were observed. The global model tests for local stochastic independence were significant in the random samples of all countries except for AT and CZ [58].

In the random samples of n = 900, item 1 (“to find out where to get professional help when you are ill”) displayed DIF for median score in FR, item 2 (“to understand information about what to do in a medical emergency”) with respect to age in PT, item 3 (“to judge the advantages and disadvantages of different treatment options”) regarding education in AT, item 6 (“to understand information about recommended health screenings or examinations”) for age in BE and CH, item 8 (“to decide how you can protect yourself from illness using information from the mass media”) for age in BE and for median score in IE, item 9 (“to find information on healthy lifestyles such as physical exercise, healthy food or nutrition”) for education in IL and for age in PT, item 10 (“to understand advice concerning your health from family or friends”) for age in CH and for median score in DE, and item 11 for age in IE [58].

Again, item 8 (“to decide how you can protect yourself from illness using information from the mass media”) has significant infit statistics in IE and SI both in the total and the random samples with values between 1.14 and 1.16, respectively. The ICC plot for SI reveals clear deviations of the observed scores from those expected by the model.

### 3.2. Distribution of the Score Values

The distributions of the type D scores are negatively (left) skewed for all countries, with a considerable ceiling effect (Figure 2), indicating that the selected HL tasks asked are manageable by many respondents. In most countries, the 75% quantile is close or equal to the maximum value of 100 (Table 11). This ceiling effect does not affect the identification of respondents with low levels of HL. Thus, the score is still sensitive for respondents with lower HL. This skewness could pose problems for some statistical analyses. The distribution of the type D score is approximately symmetric only for the German data.

The distributions of the type P scores are approximately symmetrical and almost bell-shaped in most countries (Figure 3). In some countries, the peak of the rather leptokurtic distribution shape is spiky with great excess kurtosis (e.g., PT, RU). In other countries (e.g., AT, FR, IE, IL, NO, or SI), the distribution of the type P score has an unsymmetrical shape.

### 3.3. Distribution of the Levels

Across all participating countries, about 40% of the respondents have a “sufficient” level of HL and about 15% an “excellent” level (type D, Figure 4). On the other hand, about 33% have a “problematic” level of HL and 13% an “inadequate” level.

The type P levels (Figure 5) give a less optimistic view on the general HL among the respondents of the HLS_19_ surveys. While the pattern of the extreme categories resembles the type D levels (Figure 4), the relation of problematic to sufficient levels of HL differs considerably. In general, the distribution of the four categories is less well balanced than for the type D levels with a dominance of “problematic” HL in most countries. For the PT data, this effect is amplified by the fact that respondents answered “easy“ to an above-average number of questions.

### 3.4. Validity Characteristics

#### 3.4.1. Content and Face Validity

By using the theory-based matrix of the comprehensive multifaceted model of general HL for operationalization of HLS_19_-Q12, i.e., for selection of items for the respective cells of the defining matrix, the content and face validity of the HLS_19_-Q12 is ensured.

#### 3.4.2. Discriminant Validity

Data for certain specific HL-instruments were collected only in a subset of the participating countries. With a few exceptions, most weighted Pearson correlation coefficients between the HLS_19_-Q12 score and the scores for special HL are within the target range of 0.4 to 0.7 (Table 12).

For the type D score, the value of the correlation coefficients with communicative HL is below 0.4 for AT and BE data. The correlation coefficient with vaccination HL is above 0.7 for the IT data.

For the type P score, the value of the correlation coefficient with digital HL is larger than 0.7 for the IL data, and for vaccination HL for the IT and SI data.

#### 3.4.3. Concurrent Predictive Validity—Associations with Determinants (Social Gradient)

The distribution of the selected socio-demographic and socio-economic predictor variables as examples for associations of general HL with social determinants discussed in the respective literature is described in the Appendix A.

For the model for the type D score as an outcome variable (Table 13), financial deprivation is a statistically significant predictor (*p* ≤ 0.01) in every country data but BE. Self-perceived social status in society is the second important predictor variable in most countries. All other variables (sex, age, education) are associated with significant coefficients for some of the country data, but the size of the coefficients is remarkably smaller (e.g., gender) or there is no consistent pattern across all countries: e.g., for age or education, even the sign of the regression coefficient varies. R^2^ ranges from 0.04 to 0.25 with a median value of 0.09. The residuals show some disadvantageous patterns (what is to be expected given the skewness of the distribution of the type D score values) that question the compliance with some formal assumptions of simple linear models like linearity, normality, or homoscedasticity. This hints at simple linear models not being the optimal choice for modelling type D scores for the given data of the HLS_19_ surveys.

The model for the type P score as outcome variable (Table 14) gives a very similar picture. The standardized coefficients are close to the values in Table 13 with the greatest deviation of coefficients for education for the Slovenian (β_D_ = 0.02 vs. β_P_ = 0.11) and Portuguese (β_D_ = 0.08 vs. β_P_ = 0.15) data, as well as age for the Austrian data (β_D_ = 0.08 vs. β_P_ = 0.16). In general, there is no clear trend, though the coefficients for the type P model are generally higher than those for the type D model. The R^2^ values deviate at most by 0.04 from the respective values for the type D model. Again, there is no general trend that the R^2^ for one type of model would be higher or lower than for the other model.

Despite the different distributions of the type D and P scores, the two types of models produce largely similar results.

#### 3.4.4. Concurrent Predictive Validity—Associations with Consequences

On average, this multivariable linear regression model for self-perceived health (ranging from 1 = very good to 5 = very bad) with the type D score as predictor variable (Table 15) explains 21% of the variance (varying from 11% in IE to 38% in BG). On average, the predictor with the highest absolute β value is age (varying from β = 0.08 in BE to β = 0.42 in SK) and the predictor with the second highest ß is financial deprivation (varying from −0.03 in BE to 0.22 HU). General HL (varying from β = −0.07 in SK to β = −0.22 in DK) and level in society (varying from −0.06 in RU to −0.27 in BE) are the predictors with the, on average, third highest absolute value for the β coefficient.

The model with the type P score gives comparable results (Table 16). While the values for the β coefficient of general HL are slightly higher in the majority of countries, the difference is very small in most countries and the sign of the differences is generally inconsistent.

### 3.5. Representation of the Long Form (HLS_19_-Q47)

The values of the weighted Pearson correlation coefficients are generally slightly lower for the type D scores than for the type P scores. For general HL, the correlation coefficients range from 0.897 to 0.951 for the type D scores and from 0.941 to 0.969 for the type P scores. The correlation coefficients are generally lower for the subdimensions that build on considerably fewer items (Table 17).

## 4. Discussion

### 4.1. Using Dichotomous or Polytomous Scores (Research Question 1)

In the International Report of the HLS_19_ study, the more cautious interpretation of the Likert response format in terms of type D scores was considered to avoid treating ordinal variables as interval scaled. As analyses in this article show, type D and type P scores yield comparable results for most analyses. The type D scores show unwanted patterns of the residuals, which could be interpreted as a sign that other types of models (e.g., a binomial GLM of the difficulty) could be a preferable choice. While the residuals violate the formal assumptions of simple linear models, this does not invalidate the results of such analyses, although the goodness-of-fit is sub-optimal, and the coefficient values could be higher with the choice of a more appropriate model. In general, we suggest using the type P score which also is more compatible with the procedures used in HLS-EU and which show more normally distributed scores. When the focus lies on comparing heterogenous populations or when there is a suspicion that certain subpopulations prefer different extreme response styles, the use of the type D score should be considered.

### 4.2. Psychometric Properties (Factorial Validity/Dimensionality) (Research Question 2)

With respect to the distributions of scores, the observed ceiling effect, and the relative ease of the suggested tasks of the HLS_19_-Q12 was unexpected because in the HLS-EU study, some of these tasks were deemed to be rather difficult. Future research should clarify the extent to which this is due to changes in the response categories, changes in the wording of the items, changes in the survey modalities, or actual changes in the average level of HL and the health care systems.

The proportion of a subpopulation with inadequate or limited HL levels is an important measure in the communication of survey results on HL and was also offered for HLS-EU data. The exact definition of cut-off values is to a certain extent arbitrary but was described transparently and justified. Arbitrariness is reflected in the differences between the two ways of calculating discrete levels of HL, but it is plausible that using the original polytomous response categories for difficulty leads to a heightened proportion of the difficult levels. Nevertheless, we recommend using this measure only with caution, along with an explanation of the limitations of such a categorization.

The HLS-EU study published Cronbach’s alpha coefficients only for the HLS-EU-Q47, not for the later developed HLS-EU-Q12 (i.e., the HLS-EU’s predecessor items of the HLS_19_-Q12). We calculated these values, and they range from 0.64 to 0.86 for the dichotomized items and from 0.81 to 0.92 for the polytomous items. This corresponds to the Cronbach’s alpha coefficients for the HLS_19_-Q12. The Cronbach’s alpha values and the single-factor analyses support the hypothesis that the HLS_19_-Q12 is a sufficiently unidimensional scale that is fit to measure general HL on the population level.

The Cronbach’s alpha value is below the recommended threshold of 0.7 only for the Austrian data for the dichotomized items. We still assume a sufficient degree of internal consistency for the Austrian survey for the following reasons: (1) the Cronbach’s alpha should be interpreted as a lower bound, (2) the recommended threshold should not be understood as a hard cut-off value, and (3) the Austrian survey was based on the same German version of the instrument that was also used in DE and CH.

For developing and validating the HLS-EU-Q12, PCM analyses but not CFA models were used [40]. For HLS_19_-Q12_,_ single-factor CFA models were analyzed in addition to PCM analyses. These CFA models fit the data well, which supports modelling the items of the HLS_19_-Q12 as manifest variables of a single latent variable, which on grounds of the items’ contents, can be referred to as general HL. The variation of the standardized parameters estimates is slightly lower for the polytomous items.

The psychometric analyses, especially the PCM analyses, suggest opportunities for improvement by modifying items in a future development of the HLS_19_ questionnaire. Some items display DIF for different person factors, such as age, gender, and educational background.

With regard to IRT analyses, Fischer [60] pointed out that the family of Rasch Models “… should be viewed as a guideline for the construction or improvement of tests, as an ideal to which a test should be gradually approximated, so that measurement can profit from the unique properties of the RM.” [60] Since the HLS_19_-Q12 is a self-reported, experience-based instrument for measuring and analyzing the HL of populations and not a performance-based test for HL of single individuals, fulfilling Rasch criteria is not considered a necessity for HLS_19_-Q12, although it is highly desirable to have questionnaires available that fit to an IRT model from the family of RMs. After applying a large number of powerful model tests and strict fitting rules, in line with the nature of a validation study, the null hypothesis of model fit for the HLS_19_-Q12 cannot be sustained in the 15 countries, for which data were available at the time of the IRT analyses, for both the PCM and the RM. The degree of model violations varies between the countries; for some countries, the deviations seem to be more pronounced with respect to the PCM and for others, with respect to the RM. Whereas some items display misfit and/or DIF in several countries and for more than one split criteria, some other items showed misfit or DIF only in one country and for one split criterium only. The possible reasons are manifold and include, for instance, somewhat different meanings of the items due to translation, social and cultural context, and differences in the health systems. To improve the HLS_19_-Q12 for all countries, the reasons for model violations have to be examined in more detail for each individual country based on the current results. Nevertheless, some similarities were found across countries which need to be addressed.

Item 6 (“to understand information about recommended health screenings or examinations”) displays DIF for age in 5 of the 15 countries when using the polytomous scoring and in two of the 15 countries for the dichotomous scoring. Item 9 (“to find information on healthy lifestyles such as physical exercise, healthy food or nutrition”) shows DIF with respect to educational background in 5 of the 15 countries for the polytomous scoring and in one country for the dichotomous scoring. The results for item 6 in terms of the polytomous scoring are consistent with those obtained by an independent Norwegian working group of the M-POHL consortium using somewhat different random subsamples in the individual countries and other software [59]. The item is relatively easier for the older age group in the respective countries. A possible reason could be the examples of health screenings provided in the item which could be less familiar to younger persons. Furthermore, most health screening programs (e.g., breast cancer, prostate cancer, colon cancer) are recommended from the age of 40 or 50. They are therefore generally perceived as less relevant by young people. For the other items which displayed DIF in the current validation study, the results are partly consistent with those obtained by the Norwegian working group [59].

Item 8 (“to decide how you can protect yourself from illness using information from the mass media”) shows a significant underfit in several countries both regarding the PCM and the RM, whereby in IE and SI, the infit statistic is > 1.3 for the polytomous scoring and the ICC plot for the RM for SI shows clear model deviations. The results regarding item 8 are also consistent with those obtained by the Norwegian working group [59].

In HLS_19_-data, unordered response categories were observed for two items in Austria, three items in Germany, eleven items in IE, and for all items in PT. The low endorsement rates in some of the categories could be the reason for the high number of items which were tagged as unordered in some countries [83]. No unordered response categories were observed for the HLS_19_-Q12 items in HLS-EU data from AT, BG, GR, ES, IR, NL, PL, and DE [58]. The wording of some items and the response categories (“easy” instead of “fairly easy” and “difficult” instead of “fairly difficult”) have been changed between HLS-EU and HLS_19_, which could be a possible explanation. In some languages (e.g., German), it could be more difficult to discriminate between “very difficult” and “difficult” than between “very difficult” and “fairly difficult”. For those countries with several items with unordered response categories, it is suggested to use the dichotomous scoring; since some of the answer categories are chosen very seldom, there is not much information loss and in many countries the deviation from the RM is less pronounced than from the PCM.

Survey-mode-specific analyses suggest that CAWI worked better than CATI; however, sample sizes for CATI are (too) small, especially in CH (*n* = 139) and IL (*n* = 290). However, the Norwegian working group also found some evidence that the different modes have an influence [59].

Although several deviations from the assumptions of the PCM and the RM were observed, which suggests a certain degree of multidimensionality, and some items showed poor measurement properties, from a more practice-oriented point of view, the HLS_19_-Q12 is a suitable short instrument to measure general HL on a population level to inform health policy. However, from a psychometric point of view, it should currently not be used for decisions on individual person level (e.g., to identify persons with low HL). DIF of some items furthermore poses problems for fair comparisons between subpopulation groups within a country and for comparisons between countries. Absence of DIF, however, is harder to achieve for questionnaires with verbal items compared to, e.g., tests with computational tasks [60].

These IRT analyses offer useful additional information on differences of psychometric properties of the different language and country forms of HLS_19_-Q12 and should be considered when benchmarking between selected countries. Furthermore, they offer information on potentials for improving the instrument in further rounds of M-POHL population surveys, but they do not question the appropriateness and usefulness of HLS_19_-Q12 as a self-reported HL population measure for adult national populations.

### 4.3. Validity (Research Question 3)

#### 4.3.1. Content and Face Validity

As for HLS-EU-Q12, by using the multifaceted theory-based matrix of the general HL for operationalization of HLS_19_-Q12, i.e., for selection of items for the respective cells of the defining matrix, the content and face validity of the HLS_19_-Q12 was ensured.

#### 4.3.2. Discriminant Validity

The weighted Pearson correlation coefficients between the HLS_19_-Q12 score and the scores for special health literacies are within the expected range of 0.4 to 0.7 with only few exceptions. This supports the thesis that the score for general HL measures something different than the other scores so that the scores can most likely be used independently, e.g., in regression analyses, though the scores are closely related.

#### 4.3.3. Concurrent Predictive Validity

For the regression models with general HL as outcome variable, the R^2^ values were slightly higher on average for the data of the HLS-EU study [3], where the values ranged from 0.08 to 0.29. One could question whether this could also be a consequence of the HLS-EU study relying on standardized sampling procedures across the participating countries and high-quality face-to-face interviews, while in the context of the HLS_19_ study, mostly CATI and CAWI interviews were used due to pragmatic reasons in the light of the SARS-CoV-2 pandemic. It should be noted that the models in the HLS-EU study used the HLS-EU-Q47 score, which could be another source for deviances. In general, the size of the beta coefficients, which were reported in the HLS-EU study, was smaller for the HLS_19_ data, but these are sensitive to the variance of the involved variables. At least the signs of the coefficients are comparable for important predictor variables.

For the regression model with general HL as predictor variable and self-perceived health as outcome variable, the possibility of a comparison with HLS-EU data is limited because only models that also include the Newest Vital Sign (NVS) score as predictor variable were published [10,84]. Another deviation is that in the HLS-EU models, the HLS-EU-Q47 score was used that built on the 47-items long form. Since the NVS score is of lesser importance in these published models, they are suitable references, nevertheless. The order of the β coefficients is comparable with the sole exception of financial deprivation that is slightly more prominent in the HLS_19_ data. The R^2^ coefficients were slightly higher for the published HLS-EU models.

For type D scores, there are regression analyses for further indicators of hypothesized consequences available in the International Report [12], which strengthen the concurrent predictive validity of the HLS_19_-Q12 measure.

### 4.4. Representation of Long Form HLS_19_-Q47 (Research Question 4)

The correlation of the HLS-EU-Q12 and the HLS-EU-Q47 indices was high in the original total sample of all eight countries (*r* = 0.955). In the individual countries, the correlations varied between 0.935 and 0.966 [16]. For the general HL score, the correlation of the HLS_19_-Q12 and the HLS_19_-Q47, that contains about four times the number of items, is strong enough (for D type 0.931, varying between 0.897 and 0.951; and for P type 0.958, varying between 0.941 and 0.969) to justify the use of the short form as a substitute for the long form.

For the subdimensions, the correlation coefficients are generally above 0.8 for the type P scores and above 0.75 in most countries for the type D scores, while the Cronbach’s alpha coefficients often are below 0.6 for the type D scores of the subdimensions consisting of three or four items. For the type P scores, the Cronbach’s alpha coefficients could be expected to be above 0.7. The use of the sub-scores derived from the HLS_19_-Q12 short form thus seems questionable for type D scores. If users of this instrument consider this as a necessary requirement, we suggest using the type P scores for the sub-dimensions.

### 4.5. Strengths and Limitations

With respect to the HLS-EU study and the respective instrument, the HLS_19_-Q12 instrument has the following advantages: it is available in more languages, it was tested in more countries, with a variety of data collection methods, and there was a more rigorous testing of its dimensionality. In addition, indicators for testing discriminant and concurrent predictive validity of the score showed expected results. In every HLS_19_ survey, the instrument was proven to work sufficiently well to measure the HL of general adult populations.

From a statistical perspective, the main limitation of the country data collected for the HLS_19_ study is that some items were estimated as being too easy by a majority of the respondents. The response category “very difficult” was rarely used for some items. It is unclear whether this is a problem with the items, the response categories, or the data collection method (CAWI and especially, CATI). The dichotomization of the items could be one way to avoid statistical problems arising from one extreme response category being rarely selected. It could also be a means to attenuate problems from differing extreme response preferences. It should be noted, though, that the resulting ceiling effect for the D type version could make it more difficult to measure improvements in the average level of HL following an intervention or a change of policies.

Due to difficulties in getting funding and finding suitable agencies for data collection in some countries, data were collected over an extended period of time and in different stages and contexts of the SARS-CoV-19 pandemic, which, besides other differences, affects the comparability of results. That is also true of countries using different kinds of data collection but has the advantage that the instrument is now already validated for various data collection modes.

## 5. Conclusions

For monitoring and benchmarking comprehensive, general HL in general adult populations on a national and international level, a validated internationally accepted compact measure is needed. The HLS_19_-Q12 (12 items), based on the HLS-EU-Q47, fulfils this requirement and has been validated in 17 countries and for 17 languages using different types of data collection with acceptable psychometric and validity properties. There are also indications for potential improvements of the instrument. Following the shift in measurement theory and practice [85], the HLS_19_-Q12 instrument must be validated again, when used in further studies and samples.

## 6. Use the Instrument

The ownership of the HLS_19_-Q12 rests with the HLS_19_ Consortium, which developed the instrument. The HLS_19_-Q12 can be used by third parties for research purposes free of charge but requires a contractual agreement between the user and the International Coordination Centre of the HLS_19_ Consortium. An application form with details on the conditions for getting permission to use the instrument can be found at https://m-pohl.net/tools (accessed on 1 October 2022).

## Figures and Tables

**Figure 1 ijerph-19-14129-f001:**
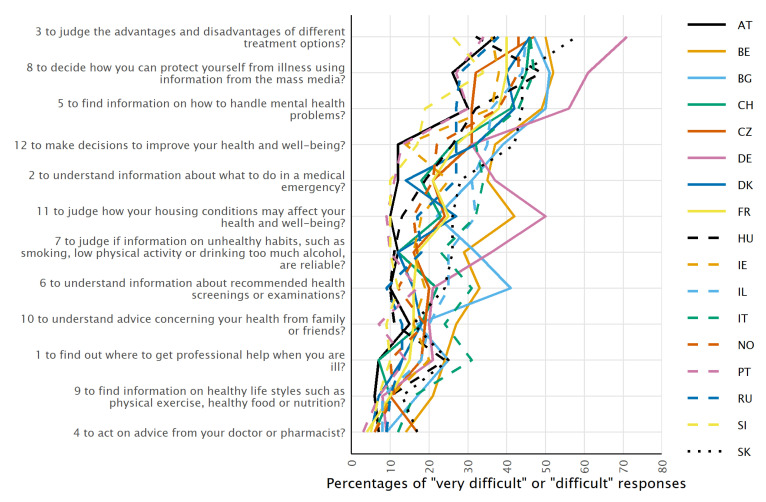
Percentages of respondents who responded with “very difficult” or “difficult” to the HLS_19_-Q12 items (ordered by the overall mean), for each country.

**Figure 2 ijerph-19-14129-f002:**
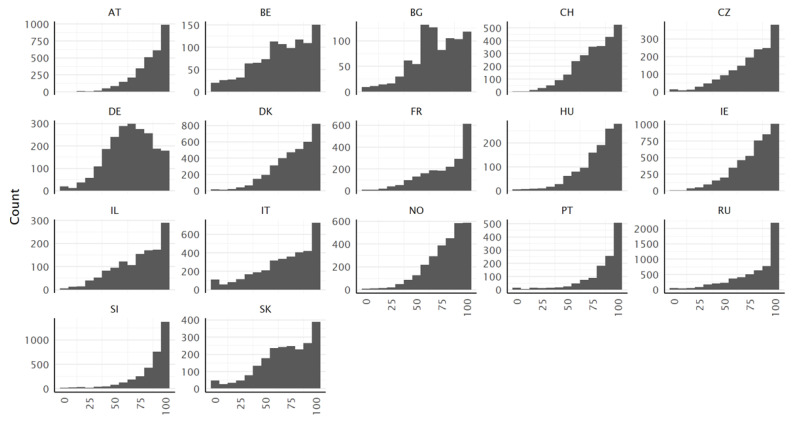
Histograms of the HLS_19_-Q12 scores (type D), for all countries.

**Figure 3 ijerph-19-14129-f003:**
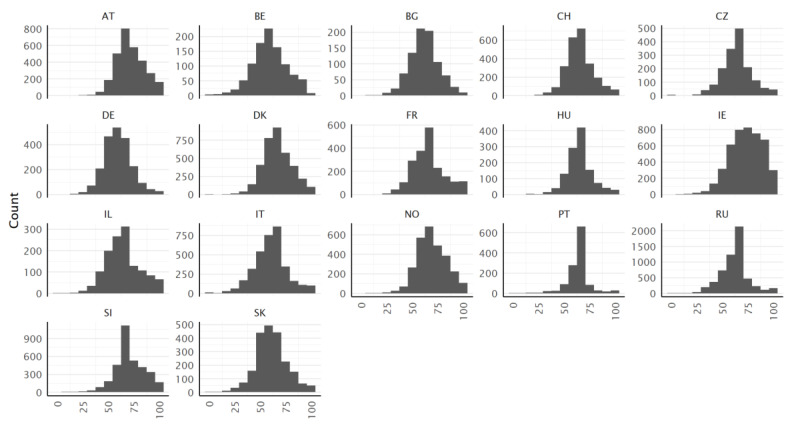
Histograms of the HLS_19_-Q12 scores (type P), for all countries.

**Figure 4 ijerph-19-14129-f004:**
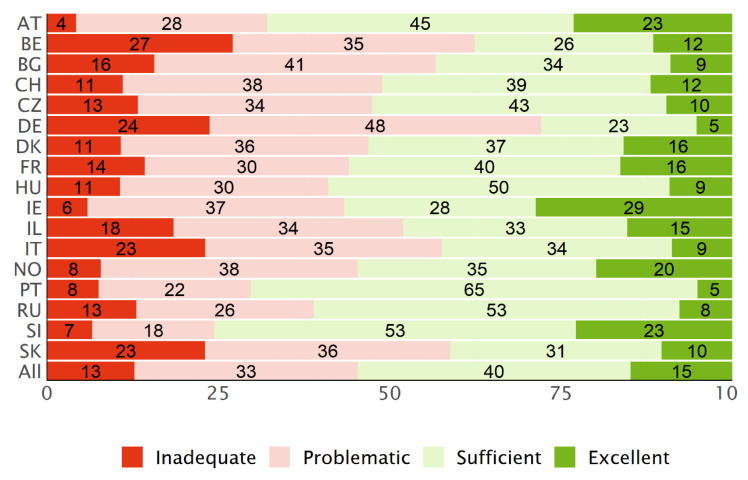
Distribution of the HLS_19_-Q12 levels for general HL, based on type D scores, for each country and the mean for all countries.

**Figure 5 ijerph-19-14129-f005:**
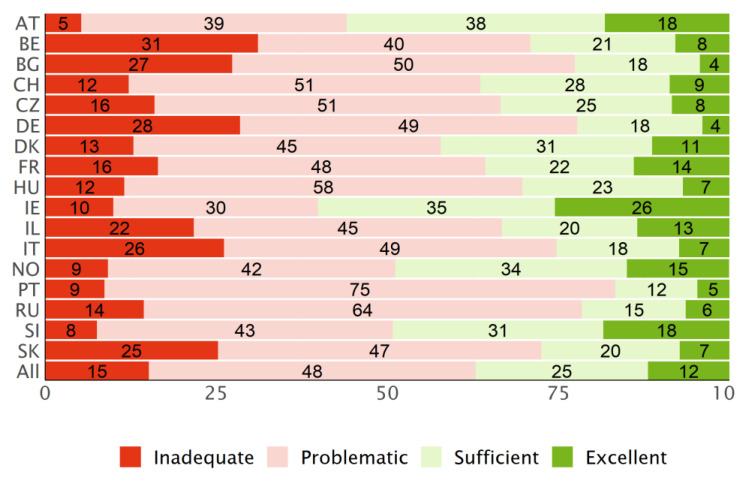
Distribution of the HLS_19_-Q12 levels for general HL, based on type P scores, for each country and the mean for all countries.

**Table 1 ijerph-19-14129-t001:** Matrix of subdimensions of HL based on the HLS-EU Conceptual Model [8] used for developing the HLS_19_ instruments.

Health Literacy	Access/Obtain Information Relevant for Health	Understand Information Relevant for Health	Appraise/Judge/Evaluate Information Relevant for Health	Apply/Use Information Relevant for Health
Health Care	(1) Ability to access information on medical or clinical issues	(2) Ability to understand medical information and derive meaning	(3) Ability to interpret and evaluate medical information	(4) Ability to make informed decisions on medical issues
Disease Prevention	(5) Ability to access information on risk factors	(6) Ability to understand information on risk factors and derive meaning	(7) Ability to interpret and evaluate information on risk factors	(8) Ability to judge the relevance of information on risk factors
Health Promotion	(9) Ability to update oneself on health issues	(10) Ability to understand health-related information and derive meaning	(11) Ability to interpret and evaluate information on health-related issues	(12) Ability to form a reflected opinion on health issues

**Table 2 ijerph-19-14129-t002:** Main characteristics of the national HLS_19_ surveys.

Country	Languages	Type of Data Collection	Sampling Procedure	Item Set	*n*	*n* for IRT ^4^
Austria (AT)	German	CATI	Multi-stage random sampling	Q12	2967	2471
Belgium (BE)	Dutch, French	CAWI	Quota sampling	Q22	1000	1000
Bulgaria (BG)	Bulgarian	CAPI, CAWI	Proportional stratified sampling and random quota sampling	Q47	865	-
Czech Republic (CZ)	Czech	CATI, CAWI	Random digital procedure and random quota sampling	Q22	1599	1459
Denmark (DK)	Danish	CAWI	Multi-stage random sampling	Q22	3602	3506
France (FR)	French	CAWI ^3^	Quota sampling	Q22	2003	2003
Germany (DE)	German	PAPI	Multi-stage random and quota sampling	Q47	2143	1991
Hungary (HU)	Hungarian	CATI	Multi-stage random sampling	Q22	1195	1021
Ireland (IE)	English	CATI	Random digit dialing approach	Q47	4487	4172
Israel (IL)	Hebrew, Arab, Russian	CATI, CAWI	Multi-stage random sampling	Q22	1315	1294
Italy (IT)	Italian	CATI, CAWI	Proportional stratified sampling	Q47	3500	-
Norway (NO)	Norwegian	CATI	Random sampling procedure within each stratum	Q47	2855	2387
Portugal (PT)	Portuguese	CATI	Random stratified sampling	Q12	1247	922
Russian Federation ^1^ (RU)	Russian	PAPI	Multi-stage random sampling	Q22	5660	4752
Slovakia (SK)	Slovak	CAPI	Multi-stage random sampling	Q22	2145	2144
Slovenia (SI)	Slovenian	CAPI, SAQ, CAWI	Multi-stage random sampling	Q47	3360	3178
Switzerland (CH)	French, German, Italian	CAWI ^2^	Multi-stage random sampling	Q12	2502	2370

Q12—The HLS_19_-Q12 short form with 12 items. Q22—A combination of the HLS_19_-Q12 and the adapted HLS_19_-Q16 short forms with 22 items. Q47—The HLS_19_-Q47 long form with 47 items. CATI—assisted telephone interview. CAWI—Computer-assisted web-based interview. CAPI—Computer-assisted personal interview. PAPI—Paper-assisted personal interview. SAQ—Self-administered questionnaire. N—Number of valid responses. IRT—Item Response Theory modelling. ^1^ In RU, respondents were selected from only three regions, Novosibirsk, Karelia, and Tatarstan. ^2^ CAWI was the main type of data collection; additionally, a small number of CATI interviews were conducted. ^3^ The data were collected in two waves. ^4^ The IRT analyses were conducted before the data collection was completed in all countries, which is why not all the data used in subsequent analyses were yet available for the IRT analyses.

**Table 3 ijerph-19-14129-t003:** Cronbach’s alpha and ordinal alpha for the HLS_19_-Q12 (polytomous and dichotomized items) for each country, and the mean for all countries (equally weighted).

	Dichotomized Items		Polytomous Items	
	Cronbach’s Alpha	Ordinal Alpha	Cronbach’s Alpha	Ordinal Alpha
AT	0.67	0.84	0.84	0.89
BE	0.82	0.91	0.88	0.91
BG	0.78	0.89	0.83	0.88
CH	0.72	0.86	0.84	0.88
CZ	0.78	0.89	0.84	0.88
DE	0.73	0.86	0.8	0.84
DK	0.75	0.89	0.86	0.9
FR	0.81	0.91	0.89	0.92
HU	0.76	0.89	0.84	0.89
IE	0.72	0.86	0.82	0.87
IL	0.8	0.9	0.88	0.91
IT	0.85	0.93	0.89	0.92
NO	0.73	0.87	0.84	0.88
PT	0.87	0.96	0.9	0.94
RU	0.86	0.94	0.9	0.93
SI	0.82	0.93	0.89	0.93
SK	0.81	0.91	0.88	0.91
Mean	0.78	0.9	0.86	0.9

**Table 4 ijerph-19-14129-t004:** Table model fit of the single-factor confirmatory factor analysis for dichotomized and the polytomous items.

	Dichotomized Items				Polytomous Items			
	SRMSR	RMSEA	CTI	TLI	SRMSR	RMSEA	CTI	TLI
AT	0.07	0.03	0.97	0.96	0.05	0.07	0.98	0.98
BE	0.08	0.05	0.98	0.97	0.05	0.06	0.99	0.99
BG	0.07	0.04	0.99	0.98	0.06	0.06	0.98	0.98
CH	0.07	0.03	0.98	0.97	0.05	0.07	0.98	0.97
CZ	0.05	0.03	0.99	0.99	0.04	0.05	0.99	0.99
DE	0.07	0.04	0.97	0.96	0.06	0.07	0.97	0.96
DK	0.06	0.03	0.98	0.98	0.05	0.06	0.99	0.98
FR	0.05	0.02	1.00	0.99	0.04	0.07	0.99	0.99
HU	0.07	0.03	0.98	0.98	0.05	0.06	0.99	0.98
IE	0.06	0.03	0.97	0.96	0.05	0.06	0.98	0.97
IL	0.06	0.03	0.99	0.99	0.05	0.06	0.99	0.99
IT	0.05	0.04	0.99	0.99	0.04	0.07	0.99	0.99
NO	0.07	0.04	0.97	0.96	0.04	0.05	0.99	0.98
PT	0.05	0.02	1.00	1.00	0.06	0.10	0.99	0.99
RU	0.05	0.04	0.99	0.99	0.05	0.07	0.99	0.99
SI	0.04	0.02	1.00	1.00	0.04	0.06	0.99	0.99
SK	0.06	0.04	0.99	0.98	0.05	0.07	0.99	0.99
Mean	0.06	0.03	0.98	0.98	0.05	0.07	0.99	0.98

SRMSR Standardized Root Mean Square Residual. RMSEA Root Mean Square Error of Approximation. CTI Comparative Fit Index. TLI Tucker–Lewis Index.

**Table 5 ijerph-19-14129-t005:** WLE and EAP reliability coefficients for the individual countries independent of data collection method.

Total Samples	AT	BE	CH	CZ	DE	DK	FR	HU	IE	IL	NO	PT	RU	SI	SK
WLE	0.83	0.88	0.84	0.85	0.81	0.85	0.88	0.84	0.79	0.87	0.83	0.84	0.88	0.87	0.88
EAP	0.85	0.88	0.84	0.85	0.80	0.86	0.89	0.85	0.82	0.89	0.84	0.85	0.89	0.89	0.89

WLE—Warm’s weighted likelihood estimate. EAP—Expected a posteriori.

**Table 6 ijerph-19-14129-t006:** WLE and EAP reliability coefficients separately for data collection method for respective countries.

By Data Collection Method	CH_CAWI_	CH_CATI_	CZ_CAWI_	CZ_CATI_	IL_CAWI_	IL_CATI_	SI_CAWI_	SI_CAPI_
WLE	0.84	0.81	0.85	0.85	0.87	0.84	0.87	0.88
EAP	0.85	0.81	0.85	0.86	0.89	0.85	0.89	0.89

**Table 7 ijerph-19-14129-t007:** Results of PCA/*t*-test procedure (proportion of significant *t*-tests and lower bound of CI) and the SRMSR statistics for the individual countries.

PCA/*t*-Test	AT	BE	CH	CZ	DE	DK	FR	HU	IE	IL	NO	PT	RU	SI	SK
Total sample	9.3 (8.2)	8.8 (7.2)	8.9 (7.8)	8.0 (6.7)	9.1 (8.0)	11.2 (10.2)	7.4 (6.3)	14.6 (12.6)	6.1 (5.4)	7.0 (5.7)	5.7 (4.9)	14.0 (11.9)	9.0 (8.2)	8.5 (7.5)	11.6 (10.3)
Random sample	7.7 (6.1)	9.3 (7.6)	11.1 (9.2)	10.4 (8.6)	11.0 (9.1)	10.8 (8.9)	8.6 (6.9)	14.0 (12.0)	6.9 (5.4)	8.3 (7.0)	2.1 (1.3)	12.4 (10.4)	6.8 (5.3)	11.3 (9.4)	12.8 (10.7)
SRMSR	0.060	0.066	0.064	0.054	0.065	0.057	0.059	0.078	0.070	0.049	0.063	0.075	0.050	0.078	0.056

PCA—Principal Component Analysis.

**Table 8 ijerph-19-14129-t008:** Results of PCA/*t*-test procedure (proportion of significant *t*-tests and lower bound of CI) and the SRMSR statistics separately for data collection method for respective countries.

By Data Collection Method	CH_CAWI_	CH_CATI_	CZ_CAWI_	CZ_CATI_	IL_CAWI_	IL_CATI_	SI_CAWI_	SI_CAPI_
PCA/*t*-test	9.6 (8.4)	16.5 (11.2)	9.2 (7.6)	11.7 (8.9)	8.8 (7.2)	7.2 (4.7)	9.0 (7.6)	9.0 (7.8)
SRMSR	0.065	0.110	0.056	0.078	0.054	0.079	0.078	0.067

**Table 9 ijerph-19-14129-t009:** Dependent item pairs.

Item Pair	Countries
1 (access, HC)—2 (understand, HC)	PT ^a,b^
6 (understand, DP)—7 (appraise, DP)	FR ^a,c^, HU ^b^, SI ^a,d^
7 (appraise, DP)—9 (access, HP)	PT ^a,b^
9 (access, HP)—10 (understand, HP)	PT ^a,b^

Notes: HC: healthcare; DP: disease prevention; HP: health promotion. ^a^: r > 0.30 in the random sample; ^b^: r > 0.30 in the total sample; ^c^: r > 0.20 in the total sample; ^d^: r > 0.25 in the total sample.

**Table 10 ijerph-19-14129-t010:** Significantly underfitting items.

	CH	CH_CAWI_	DE	FR	IE	NO	SI		SI_CAWI_	SI_CAPI_
Item	8	8	10	1	8	8	8	12	8	8
Infit	1.15	1.15	1.15	1.17	1.21	1.16	1.35	1.13	1.33	1.28
*t*	5.13	5.07	4.38	4.88	9.57 *	5.60	12.69 *	4.88	8.34	7.14

Notes: *: also significant in the random sample of *n* = 900.

**Table 11 ijerph-19-14129-t011:** Descriptive statistics of type D and type P score, for each country and for all countries (equally weighted).

	Type D Score					Type P Score				
	Mean	SD	Q25	Median	Q75	Mean	SD	Q25	Median	Q75
AT	84.8	16.1	75.0	90.9	100.0	71.8	13.6	63.3	69.7	80.6
BE	65.7	26.7	50.0	66.7	91.7	59.4	17.0	50.0	58.3	69.4
BG	68.5	23.1	54.5	66.7	90.0	59.0	13.7	50.0	58.3	66.7
CH	77.3	19.6	66.7	83.3	91.7	65.4	13.3	55.6	63.9	72.2
CZ	76.3	22.3	63.6	83.3	91.7	64.2	14.1	55.6	63.9	72.2
DE	64.9	21.9	50.0	66.7	83.3	58.9	13.6	50.0	58.3	66.7
DK	77.4	20.5	66.7	83.3	91.7	66.6	14.5	58.3	66.7	77.8
FR	77.5	22.9	58.3	83.3	100.0	65.9	15.5	55.6	63.9	75.0
HU	79.8	19.8	66.7	83.3	91.7	64.6	12.5	58.3	63.9	69.4
IE	78.8	19.5	66.7	83.3	91.7	72.3	16.3	61.1	72.2	86.1
IL	73.0	23.9	58.3	75.0	91.7	64.0	16.4	52.8	63.9	75.0
IT	69.1	27.4	50.0	75.0	91.7	60.1	16.4	50.0	61.1	69.4
NO	78.8	19.2	66.7	83.3	91.7	68.9	14.4	58.3	66.7	80.0
PT	84.8	20.5	80.0	91.7	100.0	63.8	11.5	60.0	63.9	66.7
RU	80.3	23.3	66.7	90.9	100.0	63.5	13.2	55.6	63.9	66.7
SI	86.0	19.1	83.3	91.7	100.0	70.4	14.7	63.9	66.7	80.6
SK	69.7	25.1	50.0	75.0	91.7	60.8	15.5	50.0	61.1	69.4
All	76.0	22.9	58.3	83.3	91.7	64.7	15.2	55.6	63.9	72.2

SD—Standard deviation. Q25—25%—quantile. Q75—75%—quantile.

**Table 12 ijerph-19-14129-t012:** Weighted Pearson correlation of HLS_19_-Q12_score with HLS_19_ scores for special health literacies.

	Correlation of HLS_19_-Q12 (D Type) with					Correlation of HLS_19_-Q12 (P Type) with				
	HLS_19_-COM-P- Q11	HLS_19_-COM-P- Q6	HLS_19_- DIGI	HLS_19_- NAV	HLS_19_- VAC	HLS_19_-COM-P- Q11	HLS_19_-COM-P- Q6	HLS_19_- DIGI	HLS_19_- NAV	HLS_19_- VAC
AT	0.37	0.34	0.46	0.56	0.43	0.54	0.52	0.53	0.59	0.64
BE	-	0.27	0.44	0.41	0.43	-	0.35	0.52	0.42	0.49
BG	-	0.49	-	-	0.53	-	0.61	-	-	0.54
CH	-	-	0.49	0.56	-	-	-	0.52	0.62	-
CZ	-	0.47	0.57	0.55	0.44	-	0.51	0.59	0.59	0.46
DE	0.54	0.50	0.59	0.60	0.54	0.59	0.56	0.56	0.64	0.61
DK	-	0.47	0.54	-	-	-	0.55	0.59	-	-
FR	-	0.52	0.59	0.63	-	-	0.60	0.67	0.70	-
HU	-	0.36	0.50	-	0.43	-	0.47	0.52	-	0.51
IE	-	-	0.49	-	0.55	-	-	0.58	-	0.64
IL	-	-	0.67	-	-	-	-	0.72	-	-
IT	-	-	-	-	0.72	-	-	-	-	0.79
NO	-	-	0.48	-	0.55	-	-	0.59	-	0.69
PT	-	-	0.55	0.53	0.45	-	-	0.61	0.58	0.49
RU	-	-	-	-	-	-	-	-	-	-
SI	0.48	0.45	-	0.61	0.61	0.57	0.55	-	0.67	0.73
SK	-	-	0.54	-	-	-	-	0.58	-	-
Mean	0.46	0.43	0.53	0.56	0.52	0.57	0.52	0.58	0.60	0.60

HLS_19_-COM-P-Q11—Communicative HL (with physicians in health care services) long-form score. HLS_19_-COM-P-Q6—Communicative HL (with physicians in health care services) short-form score. HLS_19_-DIGI—Digital HL score. HLS_19_-NAV—Navigational HL score. HLS_19_-VAC—Vaccination HL score.

**Table 13 ijerph-19-14129-t013:** Multivariable linear regression models of the type D score by five core social determinants (standardized coefficients (β) and R^2^), for each country and for all countries (equally weighted).

	Unstandardized						Standardized					R^2^	Valid Count	Total Count
	1	Sex	Age	Edu	Stat	Fin	Sex	Age	Edu	Stat	Fin			
AT	91.70	2.30	−0.07	−0.32	0.12	−4.26	0.07	−0.08	−0.03	0.01	−0.21	0.05	2689	2967
BE	42.26	1.21	0.05	−0.84	3.43	1.14	0.02	0.03	−0.06	0.20	0.05	0.04	985	1000
BG	52.40	1.55	−0.13	1.24	3.44	−4.32	0.03	−0.08	0.11	0.26	−0.18	0.25	724	865
CH	77.13	0.31	0.00	−0.27	0.99	−2.52	0.01	0.00	−0.03	0.08	−0.15	0.04	2009	2502
CZ	68.90	4.05	0.14	−1.66	1.73	−4.44	0.09	0.11	−0.14	0.13	−0.21	0.10	1563	1599
DE	56.91	3.10	−0.12	1.55	1.25	−3.24	0.07	−0.10	0.13	0.09	−0.15	0.09	1822	2143
DK	65.83	2.43	0.10	0.39	1.20	−5.14	0.06	0.07	0.03	0.10	−0.19	0.08	3563	3602
FR	72.55	1.85	0.00	−0.66	2.04	−3.37	0.04	0.00	−0.04	0.14	−0.15	0.06	1969	2003
HU	83.79	−1.10	0.13	0.28	0.07	−4.56	−0.03	0.11	0.03	0.01	−0.28	0.09	1122	1195
IE	70.19	2.06	0.07	0.50	1.35	−4.59	0.06	0.05	0.05	0.11	−0.24	0.10	4277	4487
IL	66.42	3.47	0.14	−1.28	1.78	−4.62	0.07	0.09	−0.09	0.14	−0.21	0.10	1154	1315
IT	81.40	2.51	−0.07	0.20	0.09	−6.47	0.05	−0.05	0.01	0.01	−0.27	0.08	3248	3500
NO	72.38	2.63	0.00	0.42	1.02	−5.33	0.07	0.00	0.04	0.08	−0.15	0.04	2675	2855
PT	91.24	−2.85	−0.19	0.82	1.70	−3.11	−0.07	−0.15	0.08	0.11	−0.18	0.15	1168	1247
RU	83.94	2.22	−0.21	0.56	2.20	−5.44	0.04	−0.15	0.03	0.16	−0.27	0.22	5012	5660
SI	91.55	1.19	−0.14	0.19	1.07	−3.07	0.03	−0.13	0.02	0.09	−0.20	0.10	3164	3360
SK	69.35	3.54	−0.16	0.45	2.50	−6.79	0.07	−0.11	0.04	0.16	−0.32	0.21	1794	2145
All	78.05	1.69	−0.05	−0.47	1.42	−4.17	0.04	−0.04	−0.04	0.10	−0.21	0.07		

1—Intercept. Sex—Gender is female (vs male as baseline category). Age—Age in years. Edu—Education in 9 ISCED levels (range 0 to 8). Stat—Self-perceived level in society (from 1 = lowest level to 10 = highest). Fin—Financial deprivation (4 levels from none to severe).

**Table 14 ijerph-19-14129-t014:** Multivariable linear regression models of the type P score by five core social determinants (standardized coefficients (β) and R^2^), for each country and for all countries (equally weighted).

	Unstandardized						Standardized					R^2^	Valid Count	Total Count
	1	Sex	Age	Edu	Stat	Fin	Sex	Age	Edu	Stat	Fin			
AT	78.82	1.87	−0.13	−0.04	0.31	−3.70	0.07	−0.16	−0.00	0.03	−0.21	0.08	2689	2967
BE	42.66	1.10	0.04	−0.40	2.22	0.55	0.03	0.04	−0.04	0.20	0.04	0.04	985	1000
BG	54.11	2.20	−0.13	0.40	1.87	−2.82	0.07	−0.14	0.06	0.22	−0.19	0.26	724	865
CH	65.38	0.70	−0.03	0.15	0.59	−1.94	0.03	−0.04	0.02	0.07	−0.17	0.05	2009	2502
CZ	56.54	2.94	0.07	−0.69	1.34	−2.56	0.10	0.08	−0.09	0.16	−0.20	0.09	1563	1599
DE	53.80	1.67	−0.09	1.05	0.80	−1.72	0.06	−0.11	0.14	0.09	−0.13	0.09	1822	2143
DK	56.63	2.36	0.03	0.39	1.04	−2.83	0.08	0.03	0.04	0.13	−0.15	0.07	3563	3602
FR	59.31	1.18	−0.01	−0.09	1.54	−1.74	0.04	−0.01	−0.01	0.15	−0.11	0.05	1969	2003
HU	64.57	−0.26	0.07	0.53	0.09	−2.58	−0.01	0.10	0.08	0.01	−0.24	0.08	1122	1195
IE	64.28	2.28	0.07	0.21	1.10	−3.63	0.07	0.07	0.03	0.11	−0.22	0.10	4277	4487
IL	58.31	3.12	0.06	−0.64	1.13	−2.66	0.09	0.06	−0.07	0.13	−0.17	0.07	1154	1315
IT	69.21	1.16	−0.10	0.49	0.03	−3.75	0.04	−0.11	0.06	0.00	−0.27	0.09	3248	3500
NO	63.00	2.58	−0.02	0.57	0.59	−3.25	0.09	−0.03	0.07	0.06	−0.12	0.04	2675	2855
PT	63.35	−1.60	−0.09	0.82	1.21	−1.45	−0.07	−0.14	0.15	0.15	−0.16	0.19	1168	1247
RU	62.12	1.65	−0.14	0.93	1.04	−2.41	0.06	−0.18	0.09	0.14	−0.22	0.18	5012	5660
SI	70.30	2.21	−0.14	0.77	0.92	−1.97	0.08	−0.17	0.11	0.10	−0.17	0.14	3164	3360
SK	57.77	2.25	−0.09	0.41	1.77	−4.07	0.07	−0.10	0.06	0.19	−0.31	0.22	1794	2145
All	63.77	1.52	−0.05	−0.01	1.07	−2.70	0.05	−0.06	−0.00	0.12	−0.20	0.08		

1—Intercept. Sex—Gender is female (vs male as baseline category). Age—Age in years. Edu—Education in 9 ISCED levels (range 0 to 8). Stat—Self-perceived level in society (from 1 = lowest level to 10 = highest). Fin—Financial deprivation (4 levels from none to severe).

**Table 15 ijerph-19-14129-t015:** Multivariable linear regression models of self-perceived health by general HL (D type) and five core social determinants (standardized coefficients (β) and R^2^), for each country and for all countries (equally weighted).

	Unstandardized							Standardized						R^2^	Valid Count	Total Count
	1	HL	Sex	Age	Edu	Stat	Fin	HL	Sex	Age	Edu	Stat	Fin			
AT	2.38	−0.01	−0.02	0.01	−0.03	−0.06	0.14	−0.18	−0.01	0.23	−0.06	−0.11	0.14	0.16	2691	2967
BE	3.56	0.00	0.07	0.00	−0.04	−0.14	−0.02	−0.14	0.05	0.08	−0.08	−0.27	−0.03	0.13	988	1000
BG	2.63	−0.01	0.02	0.01	−0.07	−0.05	0.11	−0.20	0.01	0.26	−0.19	−0.11	0.13	0.38	721	865
CH	2.41	−0.01	−0.07	0.01	−0.01	−0.08	0.10	−0.15	−0.04	0.22	−0.03	−0.18	0.15	0.16	2019	2502
CZ	2.16	0.00	−0.05	0.02	−0.06	−0.07	0.13	−0.08	−0.03	0.35	−0.11	−0.13	0.16	0.24	1567	1599
DE	1.98	−0.01	−0.03	0.02	−0.01	−0.05	0.10	−0.14	−0.02	0.41	−0.03	−0.09	0.13	0.26	1845	2143
DK	3.08	−0.01	−0.08	0.01	−0.02	−0.07	0.18	−0.22	−0.05	0.11	−0.03	−0.14	0.17	0.16	3561	3602
FR	2.60	−0.01	−0.01	0.01	0.01	−0.12	0.08	−0.16	−0.01	0.24	0.02	−0.22	0.10	0.17	2003	2003
HU	2.06	−0.01	0.08	0.02	−0.02	−0.07	0.16	−0.12	0.04	0.31	−0.05	−0.12	0.22	0.27	1124	1195
IE	2.37	0.00	−0.02	0.01	−0.04	−0.06	0.13	−0.11	−0.02	0.11	−0.10	−0.11	0.16	0.11	4301	4487
IL	1.98	0.00	−0.02	0.02	0.00	−0.08	0.10	−0.13	−0.01	0.31	−0.01	−0.16	0.12	0.17	1154	1315
IT	2.37	0.00	0.00	0.01	0.01	−0.07	0.10	−0.12	0.00	0.24	0.02	−0.14	0.16	0.14	3240	3500
NO	2.54	−0.01	0.04	0.01	−0.04	−0.09	0.25	−0.14	0.02	0.18	−0.09	−0.17	0.15	0.13	2681	2855
PT	2.00	0.00	0.13	0.02	−0.04	−0.05	0.10	−0.12	0.09	0.35	−0.11	−0.09	0.16	0.35	1168	1247
RU	2.32	0.00	0.02	0.01	−0.03	−0.02	0.10	−0.14	0.02	0.36	−0.05	−0.06	0.17	0.29	5079	5660
SI	1.90	−0.01	0.04	0.02	−0.03	−0.04	0.13	−0.15	0.02	0.37	−0.07	−0.08	0.18	0.31	3184	3360
SK	1.71	0.00	0.03	0.02	−0.02	−0.06	0.11	−0.07	0.01	0.42	−0.05	−0.11	0.16	0.33	1794	2145
All	2.39	−0.01	0.01	0.01	−0.02	−0.07	0.12	−0.15	0.01	0.26	−0.05	−0.15	0.16	0.21		

Self-perceived health—Ranging from 1 = very good to 5 = very bad. 1—Intercept. HL—HLS_19_-Q12 score. Sex—Gender is female (vs male as baseline category). Age—Age in years. Edu—Education in 9 ISCED levels (range 0 to 8). Stat—Self-perceived level in society (from 1 = lowest level to 10 = highest). Fin—Financial deprivation (4 levels from none to severe).

**Table 16 ijerph-19-14129-t016:** Multivariable linear regression models of self-perceived health by general HL (P type) and five core social determinants (standardized coefficients (β) and R^2^), for each country and for all countries (equally weighted).

	Unstandardized							Standardized						R^2^	Valid Count	Total Count
	1	HL	Sex	Age	Edu	Stat	Fin	HL	Sex	Age	Edu	Stat	Fin			
AT	2.33	−0.01	−0.02	0.01	−0.02	−0.06	0.15	−0.17	−0.01	0.22	−0.05	−0.11	0.14	0.16	2691	2967
BE	3.66	−0.01	0.08	0.00	−0.03	−0.14	−0.02	−0.14	0.05	0.08	−0.08	−0.27	−0.03	0.13	988	1000
BG	2.94	−0.01	0.03	0.01	−0.08	−0.05	0.10	−0.23	0.02	0.24	−0.20	−0.12	0.13	0.38	721	865
CH	2.60	−0.01	−0.06	0.01	−0.01	−0.08	0.10	−0.17	−0.04	0.21	−0.03	−0.18	0.15	0.16	2019	2502
CZ	2.20	−0.01	−0.05	0.02	−0.05	−0.07	0.14	−0.08	−0.03	0.35	−0.11	−0.13	0.16	0.24	1567	1599
DE	2.13	−0.01	−0.03	0.02	−0.01	−0.05	0.10	−0.14	−0.02	0.41	−0.03	−0.09	0.13	0.26	1845	2143
DK	3.22	−0.01	−0.07	0.01	−0.02	−0.06	0.19	−0.23	−0.04	0.10	−0.03	−0.14	0.17	0.16	3561	3602
FR	2.76	−0.01	−0.01	0.01	0.01	−0.11	0.09	−0.18	−0.01	0.24	0.02	−0.21	0.11	0.18	2003	2003
HU	2.24	−0.01	0.08	0.02	−0.02	−0.07	0.16	−0.14	0.05	0.31	−0.04	−0.12	0.22	0.27	1124	1195
IE	2.41	−0.01	−0.02	0.01	−0.04	−0.06	0.13	−0.11	−0.01	0.11	−0.10	−0.11	0.16	0.11	4301	4487
IL	2.23	−0.01	−0.01	0.02	0.00	−0.07	0.09	−0.18	−0.01	0.31	−0.01	−0.16	0.11	0.18	1154	1315
IT	2.50	−0.01	0.00	0.01	0.01	−0.07	0.10	−0.13	0.00	0.23	0.03	−0.14	0.16	0.14	3240	3500
NO	2.59	−0.01	0.04	0.01	−0.04	−0.09	0.25	−0.13	0.02	0.17	−0.09	−0.17	0.15	0.13	2681	2855
PT	2.02	−0.01	0.13	0.02	−0.04	−0.05	0.10	−0.10	0.09	0.35	−0.10	−0.09	0.17	0.34	1168	1247
RU	2.49	−0.01	0.03	0.01	−0.02	−0.02	0.10	−0.16	0.02	0.35	−0.04	−0.06	0.17	0.29	5079	5660
SI	1.89	−0.01	0.05	0.02	−0.02	−0.04	0.13	−0.15	0.03	0.36	−0.06	−0.08	0.19	0.31	3184	3360
SK	1.81	0.00	0.03	0.02	−0.02	−0.05	0.11	−0.08	0.02	0.42	−0.05	−0.10	0.15	0.33	1794	2145
All	2.59	−0.01	0.02	0.01	−0.02	−0.07	0.12	−0.19	0.01	0.26	−0.04	−0.14	0.16	0.21		

Self-perceived health—Ranging from 1 = very good to 5 = very bad. 1—Intercept. HL—HLS_19_-Q12 score for general HL (range 0 to 100). Sex—Gender is female (vs male as baseline category). Age—Age in years. Edu—Education in 9 ISCED levels (range 0 to 8). Stat—Self-perceived level in society (from 1 = lowest level to 10 = highest). Fin—Financial deprivation (4 levels from none to severe).

**Table 17 ijerph-19-14129-t017:** Weighted Pearson Correlation of the HLS_19_-Q12 score (D type and P type) and its subdimensions with their respective long form based on the HLS_19_-Q47.

HLS_19_-Q47 × HLS_19_-Q12	Type	BG	DE	IE	IT	NO	SI	All
General HL	D	0.928	0.920	0.903	0.951	0.897	0.928	0.931
	P	0.945	0.944	0.941	0.969	0.950	0.963	0.958
Health Care	D	0.856	0.844	0.823	0.889	0.802	0.871	0.864
	P	0.898	0.886	0.882	0.928	0.885	0.930	0.911
Disease Prevention	D	0.861	0.848	0.817	0.894	0.802	0.869	0.863
	P	0.852	0.885	0.866	0.923	0.875	0.921	0.896
Health Promotion	D	0.815	0.787	0.777	0.863	0.762	0.795	0.809
	P	0.858	0.830	0.871	0.904	0.864	0.885	0.876
Access/obtain information relevant for health	D	0.809	0.786	0.775	0.843	0.741	0.837	0.812
	P	0.849	0.847	0.847	0.891	0.840	0.903	0.871
Understand information relevant for health	D	0.761	0.778	0.772	0.865	0.788	0.801	0.806
	P	0.763	0.818	0.843	0.900	0.867	0.891	0.860
Appraise/judge/evaluate information relevant for health	D	0.876	0.843	0.762	0.874	0.765	0.809	0.847
	P	0.904	0.873	0.822	0.908	0.844	0.885	0.886
Apply/use information relevant for health	D	0.779	0.792	0.747	0.861	0.758	0.807	0.802
	P	0.826	0.830	0.830	0.898	0.849	0.877	0.864

## Data Availability

The dataset is currently available only to members of the M-POHL consortium. From December 2024, it will be made available to external researchers under a set of conditions.

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
