# Peer review of "Measuring Comprehensive, General Health Literacy in the General Adult Population: The Development and Validation of the HLS19-Q12 Instrument in Seventeen Countries"

_ijerph, 2022, doi:10.3390/ijerph192114129_

Round 1

Reviewer 1 Report

Thank you for your submission. The manuscript is interesting and well-written.

My principal concern is the comparability of the data from the participating countries. We are told that the means of administering the survey varied between countries. It is not clear how each of the countries recruited participants. Do we know that the participants are representative of the general population from each country? There should be more details of the recruitment methods.

Are references 39 and 43 the same?

Author Response

We would like to thank reviewer 1 for the helpful comments which supported improving the paper.

Point 1: My principal concern is the comparability of the data from the participating countries. We are told that the means of administering the survey varied between countries. It is not clear how each of the countries recruited participants. Do we know that the participants are representative of the general population from each country? There should be more details of the recruitment methods.

Response for Point 1:

We now provided more details on the sampling procedure by extending Table 2 with this information. In addition, we added the sentence “Data were collected based on multi-stage random sampling or quota sampling procedures in most countries (Table 2)” at page 5. Also, we added a sentence on data collection in the text on page 5: “Data collection was carried out guided by the HLS19 study protocol in the participating countries by national data collection agencies with three exceptions (BG, DK, and SK), where data collection was carried out by the HLS19 NSCs.”

Point 2: Are references 39 and 43 the same?

Response for Point 2:

Thanks, yes, they are. The duplication has been removed.

Reviewer 2 Report

I congratulate the Authors on carrying out such a demanding and logistically difficult research. This study is very innovative, and its results are very important for science, especially in the area of health literacy.

I do not make any substantive comments to the manuscript. However I have some methodological suggestions (or questions):

1. The manuscript is very long. Its volume is very large. It contains a lot of tables and / or figures. Do the journal's requirements have no limitations in this regard? I suggest that the manuscript should be divided into two articles: one describing the research methodology, the other one with the results of the research. I leave the decision to the authors.

1.1. I also suggest that some tables be moved to supplements.

2.  Some important information is missing from the methodology:

2.1. The methodology does not include a detailed description of the survey's implementation by the interviewers. How did the authors of the study make contact with research centers from 17 countries? Do the authors come from or work there, or what role did the HLS19 Consortium of the WHO Action Network M-POHL play in the study?

2.2. What was the study like at the level of individual research centers? How did the researchers choose the interviewers? How did the researchers choose the respondents? What was the selection of the research group?

2.3. There is no description of how the data was collected? Were the datasets in each country the same? How were the results analyzed jointly for all countries, since the questionnaires were in different languages?

3. In the results section, I suggest moving some tables to supplements.

4. The discussion is a commentary on the results because a lot of the discussion elements take place in the methodology. There is little reference to other studies in the discussion itself. By contrast, they are in methodology. I suggest that these elements move the methodology into the discussion.

5. technical issues:

5.1. The each abbreviation should be expanded the first time you use it

5.2. incorrect numbering (punches) at the beginning of lines: 452-462, 467-470, 561-565, 573-574.

Author Response

We would like to thank reviewer 2 for the helpful comments which supported improving the paper.

Point 1. The manuscript is very long. Its volume is very large. It contains a lot of tables and / or figures. Do the journal's requirements have no limitations in this regard? I suggest that the manuscript should be divided into two articles: one describing the research methodology, the other one with the results of the research. I leave the decision to the authors.

Response for Point 1:

Thank you for the consideration of dividing the articles into two articles. This was discussed between the co-authors, and we would like to stick to one article. As for the length, we would like to be transparent on our research methodology and results and therefore provide extensive information. We did shorten the article by removing the tables from the appendix and instead providing these in the supplementary material.

1.1 I also suggest that some tables be moved to supplements.

Response for Point 1.1:

We did shorten the article by removing the tables from the appendix and instead providing these in the supplementary material.

  1. Some important information is missing from the methodology:

2.1. The methodology does not include a detailed description of the survey's implementation by the interviewers. How did the authors of the study make contact with research centers from 17 countries? Do the authors come from or work there, or what role did the HLS19 Consortium of the WHO Action Network M-POHL play in the study?

Response for Point 2.1:

We now provide a more detailed description of the implementation of the survey in the 17 countries and the organisational structure for conducting the research by adding the following sentences:

Page 2 “An International Coordination Center was established to provide a study protocol and enable international coordination. In each of the 17 participating countries in the WHO European Region a National Study Center (NSC) was contracted to conduct the HLS19 project:”

All authors of the article come either from the International Coordination Center or from the National Study Centers. Here we also reference the HLS19 International Report in which extensive information is provided on the HLS19 Consortium and where all names are listed.

Page 5: “Data collection was carried out guided by the HLS19 study protocol in the participating countries by national data collection agencies with three exceptions (BG, DK, and SK), where data collection was carried out by the HLS19 NSCs.”

2.2. What was the study like at the level of individual research centers? How did the researchers choose the interviewers? How did the researchers choose the respondents? What was the selection of the research group?

Response for Point 2.2:

We now provided more details on the sampling procedure by extending Table 2 with this information. In addition, we added the sentence “Data were collected based on multi-stage random sampling or quota sampling procedures in most countries (Table 2)” and “Data collection was carried out guided by the HLS19 study protocol in the participating countries by national data collection agencies with three exceptions (BG, DK and SK), where data collection was carried out by the HLS19 NSCs.” at page 5.

As for the selection of the research group this is now explained by adding the sentence on page 2: “An International Coordination Center was established to provide a study protocol and enable international coordination. In each of the 17 participating countries in the WHO European Region a National Study Center (NSC) was contracted to conduct the HLS19 project:” Here we also reference the HLS19 International Report in which extensive information is provided on the HLS19 Consortium.

2.3. There is no description of how the data was collected? Were the datasets in each country the same? How were the results analyzed jointly for all countries, since the questionnaires were in different languages?

Response for Point 2.3:

We added a sentence on data collection and on the study protocol used in all countries in the text on page 5: “Data collection was carried out guided by the HLS19 study protocol in the participating countries by national data collection agencies with three exceptions (BG, DK, and SK), where data collection was carried out by the HLS19 NSCs.”

Concerning the data sets and the analyzing of the results we added: “The data sets were submitted to the HLS19 International Coordination Center for data control and creating an English language international data template file for further analysis. Post-stratification weights were applied to improve the estimation of population parameters. For most data sets, the weights were calculated by the HLS19 NSCs to best fit the sampling procedure [12].” at page 5 and extended a sentence to “ …most analyses were done by the International Coordination Center by country. In some analyses, a reference value (e.g., a mean across all countries weighted equally) is given to facilitate interpretation.” at page 6.

  1. In the results section, I suggest moving some tables to supplements.

Response for Point 3:

We moved three tables from the appendix to the supplementary material.

  1. The discussion is a commentary on the results because a lot of the discussion elements take place in the methodology.There is little reference to other studies in the discussion itself.By contrast, they are in methodology. I suggest that these elements move the methodology into the discussion.

Response for Point 4:

We moved the first paragraph of chapter 2.3.2. to the discussion section (page 27): “With regard to IRT analyses Fischer [60] pointed out, that the family of Rasch Models: “… should be viewed as a guideline for the construction or improvement of tests, as an ideal to which a test should be gradually approximated, so that measurement can profit from the unique properties of the RM." [60] Since the HLS19-Q12 is a self-reported, experience-based instrument for measuring and analyzing HL of populations and not a performance-based test for HL of single individuals, fulfilling Rasch criteria is not considered a necessity for HLS19-Q12, although it is highly desirable to have questionnaires avail-able that fit to an IRT model from the family of RMs.”

  1. technical issues:

5.1. The each abbreviation should be expanded the first time you use it

Response for Point 5.1

This was checked and either the full terminology (instead of abbreviations) was used or abbreviations were explained.

5.2. incorrect numbering (punches) at the beginning of lines: 452-462, 467-470, 561-565, 573-574.

Response for Point 5.2

Bullets are now used.